# Suppress Content Shift: Better Diffusion Features via Off-the-Shelf Generation Techniques

**Benyuan Meng**[1,2]    **Qianqian Xu**[3,4*]    **Zitai Wang**[3]
**Zhiyong Yang**[5]    **Xiaochun Cao**[6]    **Qingming Huang**[5,3,7*]

[1]Institute of Information Engineering, CAS
[2]School of Cyber Security, University of Chinese Academy of Sciences
[3]Key Lab. of Intelligent Information Processing, Institute of Computing Technology, CAS
[4]Peng Cheng Laboratory
[5]School of Computer Science and Tech., University of Chinese Academy of Sciences
[6]School of Cyber Science and Tech., Shenzhen Campus of Sun Yat-sen University
[7]Key Laboratory of Big Data Mining and Knowledge Management, CAS
mengbenyuan@iie.ac.cn  {xuqianqian,wangzitai}@ict.ac.cn
{yangzhiyong21,qmhuang}@ucas.ac.cn  caoxiaochun@mail.sysu.edu.cn

## Abstract

Diffusion models are powerful generative models, and this capability can also be applied to discrimination. The inner activations of a pre-trained diffusion model can serve as features for discriminative tasks, namely, diffusion feature. We discover that diffusion feature has been hindered by a hidden yet universal phenomenon that we call content shift. To be specific, there are content differences between features and the input image, such as the exact shape of a certain object. We locate the cause of content shift as one inherent characteristic of diffusion models, which suggests the broad existence of this phenomenon in diffusion feature. Further empirical study also indicates that its negative impact is not negligible even when content shift is not visually perceivable. Hence, we propose to suppress content shift to enhance the overall quality of diffusion features. Specifically, content shift is related to the information drift during the process of recovering an image from the noisy input, pointing out the possibility of turning off-the-shelf generation techniques into tools for content shift suppression. We further propose a practical guideline named GATE to efficiently evaluate the potential benefit of a technique and provide an implementation of our methodology. Despite the simplicity, the proposed approach has achieved superior results on various tasks and datasets, validating its potential as a generic booster for diffusion features. Our code is available at [this url](#).

## 1   Introduction

Diffusion models (DMs) [15, 34] are a prevalent family of generative models for various tasks [36, 35, 25]. This strong generative capability can be applied to discrimination [21]. Diffusion Feature (DF), a popular approach, extracts inner activations from a pre-trained diffusion model as vision features [1, 50, 60, 57, 37, 62], similarly to how ResNet [14] serves as a feature extractor. Extracting features with a vastly pre-trained diffusion model grants this approach strong robustness and generalizability. Furthermore, it enjoys the philosophy of hitchhiking, a solid research paradigm in the age of large base models [44]: advancements in diffusion models can all be transformed into better feature quality. It promises this research direction a bright future.

---

[*]Corresponding authors.

38th Conference on Neural Information Processing Systems (NeurIPS 2024).

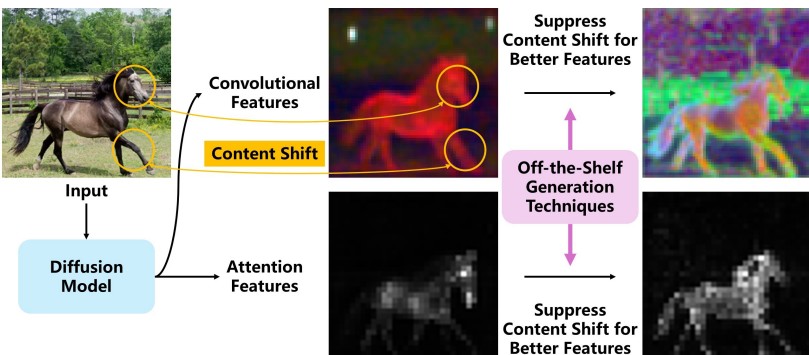

Figure 1: Current diffusion features widely suffer from content shift, *i.e.*, content differences between inputs and features. Due to the inherent connection, content shift can be suppressed with off-the-shelf generation techniques.

Have we already obtained satisfying diffusion features? The original role of diffusion models, generation, has provided an inspiring perspective. There is an endless pursuit among AIGC players[2] for better control over generation [17, 58, 53, 28]. This implies the inherent difficulty in controlling diffusion models: their generation results may not be exactly the same as intended. Will this property of diffusion models also affect the quality of diffusion features? We visualize some diffusion features in Figure 1 and find that they do contain detail differences from inputs, which might hinder the performance. We name this phenomenon *content shift*, *i.e.*, content differences between diffusion features and the input image. Although we have just shown an example with very obvious content shift, it is in fact intentionally amplified for observation. Under more practical feature extraction settings, the magnitude of content shift would be significantly weaker for visual perception. In Section 4, however, empirical results show that even in such cases, the negative impact of content shift on discrimination is not negligible, indicating the necessity to suppress it for diffusion features of better quality.

In pursuit of a method to suppress content shift, we need to further investigate why this phenomenon exists. We notice in Section 4 that the diffusion backbone reconstructs clean inner representations from noisy inputs in the middle of UNet before predicting noises based on the reconstructed content. The diffusion features we are using are in fact the reconstructed representations, which answers why we can obtain clean features from noisy images. However, since high-frequency details are potentially blurred out by noises and then recovered by "imagination", this reconstruction process inherently suffers from the risk of drifting from the original image. Content shift in diffusion features, naturally, is the reflection of such drift during reconstruction. Consequently, to suppress content shift, we need an additional way to directly introduce the original clean image into the reconstruction process and steer it towards the original image. To our delight, we notice that many off-the-shelf image generation techniques for diffusion models [58, 28, 53] also work by injecting additional information into UNet and thus steering the reconstruction. Through careful evaluation of the effect of generation techniques, we are able to select some techniques that can directly satisfy the goal of suppressing content shift, which eases the implementation and extension of our method. To guide the efficient evaluation of generation techniques, we also propose a guideline in Section 5. This method is denoted as *GenerAtion Techniques Enhanced* (**GATE**) diffusion feature and its effect is also shown in Figure 1.

To validate GATE, we implement it (Section 5) by choosing three techniques from a few highly popular generation techniques. Since the integrated techniques can benefit from feature amalgamation [24, 62] more than previous approaches, this method is also adopted in our implementation. Although this is a very simple implementation, it achieves impressive performance on various tasks and datasets (Section 6). It demonstrates the effectiveness and potential of GATE, as more techniques can still be evaluated, and there will be even more as diffusion models develop.

In summary, the contribution of this work is as follows:

---

[2]https://civitai.com/

- To the best of our knowledge, we are the first to reveal and systematically analyze the universal, harmful, yet hidden phenomenon, content shift, in diffusion features.

- We point out the possibility of utilizing off-the-shelf generation techniques for content shift suppression and propose the GATE guideline to facilitate technique integration.

- Comprehensive experiments on two discriminative tasks validate the effectiveness of our method.

## 2 Related Work

The introduction to diffusion models can be found in [6]. In this section, we solely focus on diffusion features. So far, the research topics in this direction fall into two categories: **task exploration** and **method improvement**, *i.e.*, applying the paradigm to different tasks and studying the approach itself for better feature quality, respectively.

**Task exploration.** There have been attempts on semantic segmentation [1, 50], semantic correspondence [57, 37, 22], hyperspectral image classification [62], domain generalization [9], training data synthesizing [45, 55], zero-shot referring image segmentation [30], visual grounding [23], a novel task involving personalization [47], co-salient object detection [48], and open-world segmentation [38]. This extensive and extending list shows the discriminative capability of DMs.

**Method improvement.** The way to enhance diffusion features can be divided according to the three important factors of a diffusion model: prompt, layer, and timestep. In a basic pipeline, the prompt is manually designed and simple, the layer only refers to the activations between convolutional blocks, and the timesteps are manually set. (i) To enhance the use of prompt, a typical improvement is prompt tuning [50, 60, 57, 22], which equivalently fine-tunes the text encoder along with the downstream discriminative task. Another novel method is auto-captioning [19], replacing manual prompt design with an automatic captioner. (ii) To dig more information out of UNet layers, researchers choose to additionally take attention layers into consideration. However, we find that cross-attention is more frequently used [50, 60, 57, 30] while self-attention is less popular [47]. (iii) As for better usage of timesteps, [62] proposes to extract features from many timesteps and dynamically assign weights to them, while [52] employs reinforcement learning for better timestep selection.

Of the two directions, our work aims for better methods instead of new tasks. Furthermore, most existing diffusion feature approaches suffer from content shift, an example of which is the visualization of attention features in Figure 1. Therefore, our GATE can serve as a generic performance booster to other diffusion feature approaches, showing its potential for broad application.

## 3 Preliminaries: Diffusion Feature

Diffusion models consist of a neural network module and a diffusion scheduler. The network is an end-to-end network, which can be formally denoted as $\epsilon_\theta$, where $\theta$ is the parameters. The diffusion scheduler is the core of diffusion models. With the scheduler, diffusion models generate images progressively, during which the network module is **re-used** on each timestep, each time only predicting an incremental noise. Generally, we use a **smaller / larger timestep** to indicate **less / more noises**. A typical generation process starts from $t = T$ (total noises) and ends at $t = 0$ (clean images).

Next, we will explain how features are extracted using a common pipeline for diffusion features, with the visual illustration in Figure 2. Given an input image $x \in \mathbb{R}^{3 \times h \times w}$, where $h, w$ are height and width, the extraction process includes: (i) A pre-trained VAE encodes the input image into the latent space, inducing $x_0 \in \mathbb{R}^{4 \times h' \times w'}$, as a common practice presented in [34, 33]. (ii) $x_0$ acts as the input of the forward diffusion process, *i.e.*, timestep $t = 0$. As suggested in [1], it is beneficial to extract diffusion features at non-zero timestep. Following this practice, we set $t = 50$ and get $x_t$. (iii) $x_t$, along with the timestep $t$ and a textual prompt $c$, is sent into the pre-trained diffusion UNet $\epsilon_\theta$, *i.e.*, $\epsilon_\theta(x_t, t, c)$. The generation techniques selected by our method are also applied here to modify $\theta, c$. (iv) Convolutional and attention features are gathered during the computation of the backbone. For convolutional features, we gather the output activations of each resolution in the upsampling stage of the UNet. For attention features, we obtain the mean value of the similarity maps between query and key in all cross-attention layers.

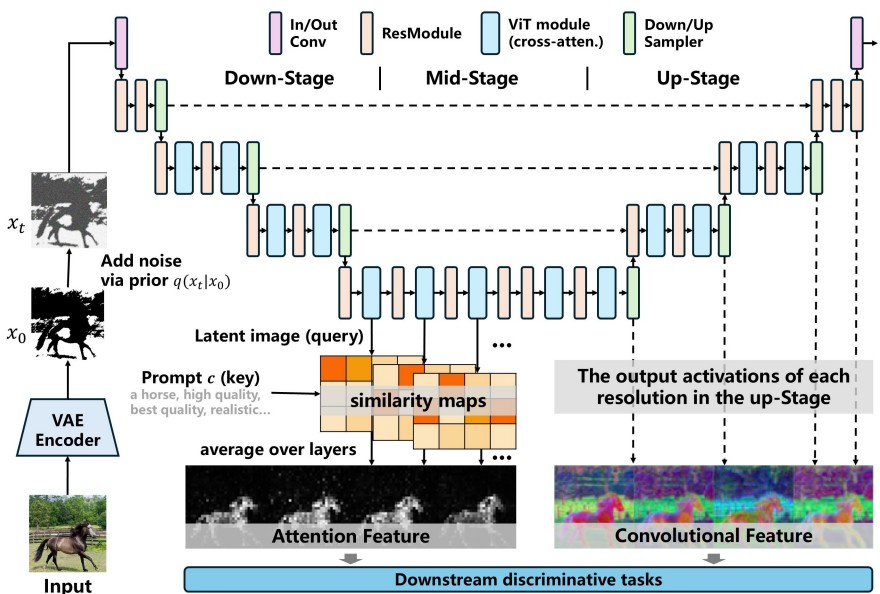

Figure 2: The overall process of feature extraction. The original image is first processed into UNet inputs via VAE and noise addition. Afterward, we collect the output activations of each resolution in the upsampling stage as convolutional features. At the same time, the cross-attention layers of UNet produce similarity maps, which are averaged over all upsampling layers as attention features.

| Dataset | Low Quality | Neutral | High Quality |
|---|---|---|---|
| Horse-21 | 58.90 | 59.03 | 59.33 |
| CIFAR10 | 91.67 | 91.44 | 91.21 |

Figure 3: The averaged results over three repeats with quality prompts. Horse-21 (**high quality**) and CIFAR10 (**low quality**) benefit from prompts closer to the image quality, suggesting the negative effect of content shift at small timesteps.

## 4 Exploration of Content Shift

### 4.1 Impact of Content Shift

In Figure 1, we can qualitatively observe content shift from feature visualization. However, this visualization is intentionally amplified for better observation by extracting features at large timesteps. Now we aim to examine the impact of content shift on quantitative performance under more practical scenarios, *i.e.*, when timesteps are small. To this end, we need to toggle the magnitude of content shift in features. Describing the quality of the image is a widely adopted way to control generation. A prompt accurately describing the quality of input images is considered to suppress content shift, while intentionally describing something different from the image will cause more severe content shift. We select the semantic segmentation task [1] as an example of high-quality images and image classification on CIFAR10 [20] for low-quality images. The results in Figure 3 seem counter-intuitive at first glance, as low-quality prompts surprisingly can improve the performance on CIFAR10, but this in fact complies with the quality of input images. Therefore, these results can demonstrate the negative impact of content shift on feature quality at small timesteps.

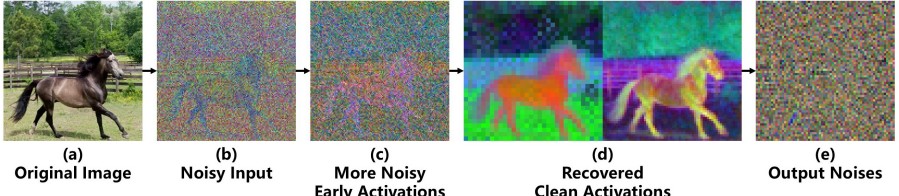

| (a) | (b) | (c) | (d) | (e) |
|-----|-----|-----|-----|-----|
| Original Image | Noisy Input | More Noisy Early Activations | Recovered Clean Activations | Output Noises |

Figure 4: Content shift is caused by the reconstruction process within diffusion model activations. This visualization consists of activations obtained during a single network forward pass.

## 4.2 Cause of Content Shift

After the negative impact of content shift is confirmed, we next aim to find its cause. Unlike more conventional feature extractors such as ResNet [14], the inputs to diffusion models are not the original image (Figure 4(a)), but its noisy version (Figure 4(b)), as enforced by the diffusion process [15]. The early layers of diffusion models even further add more noise to the feature maps (Figure 4(c)). However, the diffusion UNet gains the ability from vast pre-training to reconstruct clean inner representations from noisy inputs (Figure 4(d)), roughly at the middle of the UNet structure. Additionally, the shortcut structures in UNet also help the reconstruction by passing some high-frequency details. Afterward, the diffusion UNet will further predict noises based on the reconstructed representations (Figure 4(e)).

Notably, the diffusion features we are using are in fact the reconstructed representations, which answers why clean diffusion features can be obtained even though the inputs are noisy. Despite the reconstruction ability, however, many high-frequency details are potentially blurred out by input noises, and thus their reconstruction is mostly based on "imagination". This leads to possible drift from the original image during reconstruction [7]. Naturally, the content shift phenomenon in extracted diffusion features reflects the drift during reconstruction. **Consequently, content shift is an inherent characteristic of diffusion models and diffusion features, which suggests its broad existence across models and timesteps**.

## 5 Suppression of Content Shift

### 5.1 Utilization of Generation Techniques

According to the cause of content shift, we need to steer the reconstruction process back to the original image to suppress content shift. While it is viable to design new methods for this purpose, we find it also possible and more efficient to adopt off-the-shelf generation techniques. Specifically, as an inherent characteristic of diffusion models, content shift affects not only features but also the original generative purpose of diffusion models. Hence, there have been techniques for generation that are already capable of toggling content shift by steering reconstruction [7, 58]. For example, ControlNet [58] introduces an additional reference image and steers reconstruction by directly modulating activations, pushing the reconstructed representation towards the reference image. It inspires us to utilize ControlNet in a different way: we can set the same input image simultaneously as the reference image, enforcing the recovered image to be more similar to the input one and thus suppressing content shift. Similarly, it is possible to adopt other generation techniques to suppress content shift.

Furthermore, utilizing off-the-shelf generation techniques is more consistent with the intuition of diffusion feature than designing a new method. Specifically, this field is dependent on the generative capability of diffusion models, and thus it is important to stay updated with new advancements in the generation field. Utilizing techniques from the generation field helps with this goal, while a method that is newly designed solely for diffusion feature might hinder it. Considering it, we decide not to devise new methods, but to develop more detailed guidelines for suitably integrating these off-the-shelf generation techniques.

We next illustrate the guidelines for utilizing off-the-shelf generation techniques. In the field of generation, the abundant techniques may not all be suitable for suppressing content shift, suggesting the necessity of examining the effect of a given technique on feature quality. Although it is possible

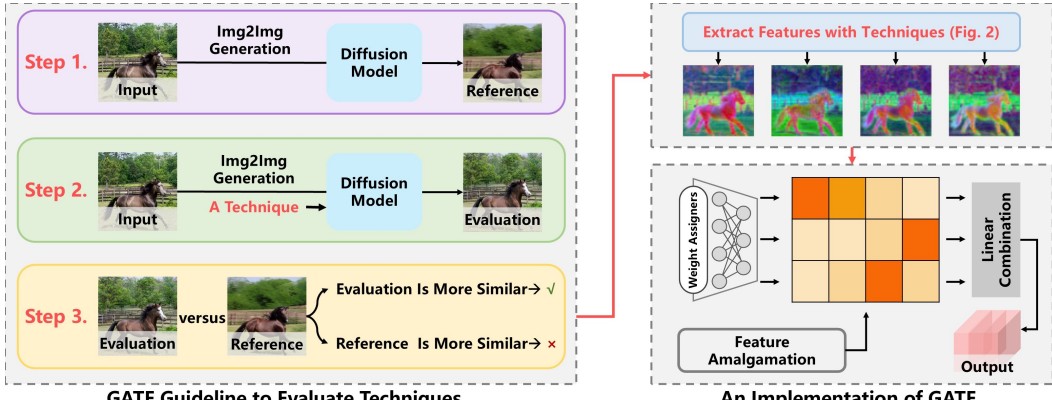

Figure 5: Overview of the GATE guideline and our implementation. GATE evaluates if a technique can suppress content shift based on the result of Img2Img generation. If a technique makes the result more similar to the input, it is considered to be potentially helpful. We further implement GATE by choosing three off-the-shelf generation techniques and amalgamating features obtained with different combinations of the techniques.

to make empirical and quantitative examinations on discriminative tasks, a more efficient evaluation protocol would be preferable. To this end, we propose the GATE guideline for technique evaluation, which is presented in Figure 5 along with other components of the framework. To avoid having to integrate a technique into the feature extraction pipeline before knowing its potential, we utilize Img2Img generation as an alternative for evaluation:

(i) Gather an Img2Img generation result as the reference image, done with high repainting strength to amplify content divergence for observation.

(ii) Apply the technique to be evaluated and perform another Img2Img generation. If the technique requires parameters, they should be set such that the output could be more similar to the input.

(iii) If the new output is more similar to the input than the reference image, the technique is considered able to suppress content shift in diffusion features.

### 5.2 Quantitative Evaluation

We have also developed a quantitative metric for evaluating generation techniques. We set feature extracted at $t = 0$ as reference $FEAT_{ref} \in \mathbb{R}^{c \times h \times w}$ as it is less affected by noises. Then we use the Laplacian operator to evaluate the contour difference between feature $FEAT$ and the reference:

$$diff = \sum_{i,j}^{h \times w} | \, \|Laplacian(FEAT_{ref}, i, j)\|_2 - \|Laplacian(FEAT, i, j)\|_2 \, | \in (-\infty, 1] \quad (1)$$

We then set a feature with stronger content shift as an anchor and compare $diff_{anchor}$ with other features.

$$Score = \frac{(diff_{anchor} - diff)}{diff_{anchor}} \quad (2)$$

$Score = 1$ means an exact match, and a smaller value indicates more shift. In this way, we can measure the extent of content shift in extracted features using different generation techniques and thus evaluate the suppression effect of techniques. Noticeably, the evaluation of this quantitative metric can be well approximated by the previously proposed qualitative evaluation, so we recommend the qualitative evaluation if efficiency is desired.

### 5.3 Selected Generation Techniques

We select three generation techniques as our implementation of GATE. Their Img2Img generation results are provided in Figure 6. Integrating the techniques only slightly impacts efficiency, which will be discussed in Appendix A along with technique details. Additionally, we analyze two failed

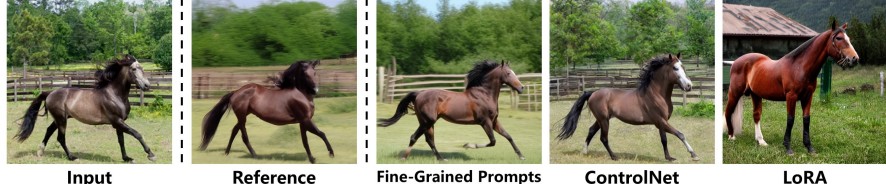

| Input | Reference | Fine-Grained Prompts | ControlNet | LoRA |

Figure 6: Img2Img generation results according to the GATE guideline, validating the potential of the three selected techniques for suppressing content shift.

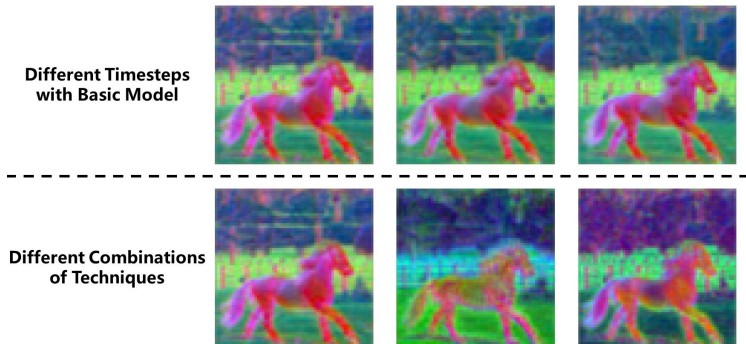

**Different Timesteps with Basic Model**

**Different Combinations of Techniques**

Figure 7: The first row shows features extracted at different timesteps. The second row is from different combinations of generation techniques and shows stronger diversity.

techniques in Appendix B, which might help better understand how generation techniques influence content shift.

**Fine-Grained Prompts.** Prompts are a description of expected image content in natural language. For example, "a single horse running in a sports field, with a well-equipped rider on its back, high quality, highly realistic, masterpiece". Modern diffusion models are inherently trained to generate images conditioned on the given prompts [34]. Hence, by describing the content of the input image in prompts, it is possible to steer the reconstruction to stay close to the input. The result of fine-grained prompts in Figure 6 is not significantly better than the reference, but this technique enjoys the advantage of being a built-in function of diffusion models, requiring no code modification.

**ControlNet.** Quite often, the control via prompts is too ambiguous. This is when ControlNet [58] can be helpful. ControlNet is a plug-in module for diffusion models, designed to take an additional control image input and push the reconstructed representation toward the control image. ControlNet presents an outstanding control effect over generation among the three techniques, as shown in Figure 6, implying its potential for suppressing content shift.

**LoRA.** LoRA [17] is an efficient replacement for model fine-tuning, which bears a similar effect to fine-tuning, thus showing to be highly effective for capturing image styles. By injecting additional knowledge of image styles of the given dataset into model weights, LoRA enables reconstruction to end with stronger similarity to the input image. From Figure 6, we can observe that although LoRA does not provide as strong a control effect as ControlNet, it can significantly promote image quality, which is believed to bring some unique advantages.

### 5.4 Feature Amalgamation

The three techniques above are able to improve feature quality individually and can be applied simultaneously for stronger suppression effects. While this can lead to a single high-quality feature, it is a common practice in previous diffusion feature approaches to amalgamate multiple features for further improvement [1, 24, 62, 29]. The conventional way extracts features at different timesteps to obtain more diverse information. We find that, compared to the conventional amalgamation of timesteps, the amalgamation of different combinations of generation techniques can bring stronger diversity, as indicated in Figure 7. Therefore, we additionally amalgamate features obtained with

various generation technique combinations to harness further enhancement. More details of how feature amalgamation is implemented are provided in Appendix A.

# 6 Experimental Validation

## 6.1 Experimental Settings

**Task & Dataset.** Typically, diffusion feature studies [1, 50, 60] prefer fine-grained pixel-level tasks for evaluation. Following this practice, we select three tasks for experiments: semantic correspondence using SPair-71k [26] dataset, label-scarce semantic segmentation using Bedroom-28 [54] and Horse-21 [54] datasets, and standard semantic segmentation using ADE20K [61] and CityScapes [5] datasets.

**Evaluation Metrics.** (i) For semantic correspondence, PCK@$0.1_{\text{img}}(\uparrow)$ and PCK@$0.1_{\text{bbox}}(\uparrow)$ are used, following the widely-adopted protocol reported in [26]. (We omit @0.1 to save some space in Table 1.) These two metrics mean the percentage of correctly predicted keypoints, where a predicted keypoint is considered to be correct if it lies within the neighborhood of the corresponding annotation with a radius of $0.1 \times max(h, w)$. For PCK@$0.1_{\text{img}}$/PCK@$0.1_{\text{bbox}}$, $h, w$ denote the dimension of the entire image/object bounding box, respectively. (ii) For semantic segmentation, we use mIoU metric, which is the mean over the IoU performance across all semantic classes [12]. For each image, IoU (Intersection over Union, $\uparrow$) is defined by #(overlapped pixels between the prediction and the ground truth) / #(union pixels of them). In addition, we also use aAcc and mAcc, where aAcc is the classification accuracy of all pixels and mAcc averages the accuracy over categories.

**Feature Extraction.** All tasks extract features at $t = 50$. When ControlNet is applied, except for standard semantic segmentation, we additionally start multi-step denoising from $t = 60$. For feature amalgamation, we extract multiple convolutional features and one attention feature per image:

  (i) Semantic correspondence: We obtain six in total convolutional features using individual fine-grained prompt, ControlNet, and LoRA techniques, and one attention feature using a prompt including all object categories, with ControlNet and LoRA.

 (ii) Label-scarce semantic segmentation: We obtain one convolutional feature using fine-grained prompts, one convolutional feature using ControlNet, and one (Bedroom-28) to two (Horse-21) features using different LoRA weights. One attention feature is extracted with all three techniques applied.

(iii) Standard semantic segmentation: One convolutional feature is obtained using only fine-grained prompts and two more are extracted additionally with ControlNet and different LoRA weights. One attention feature is extracted with all three techniques applied.

Notably, the ADE20K dataset for standard semantic segmentation contains images of varying scenes, which can test how well the fine-grained prompt technique can generalize in this scenario. To this end, we use a prompt that can cover different scenarios: "a highly realistic photo of the real world. It can be an indoor scene, or an outdoor scene, or a photo of nature. high quality". This prompt covers various scenes for generalizability and describes image quality for fine-grained effect.

For more experimental settings, including more detailed feature extraction methods and implementation details, please refer to Appendix C.

## 6.2 Comparison with SOTA

The experimental results are shown in Table 1 and Table 2. For most SOTA competitors, we borrow the reported results from their original studies. However, MaskCLIP [8] and ODISE [50] only provide results on ADE20K and it is hard to extend their implementations to CityScapes, so their results on CityScapes are missing. Furthermore, the original results reported by VPD [60] are based on full-scale fine-tuning of diffusion UNet, which is not fair as we do not train the diffusion model. Therefore, we re-evaluated VPD with the diffusion UNet frozen and reported our results.

**Semantic Correspondence.** It is a pity that the related studies do not perform experiments under exactly the same setting. For fairness, we mainly compare GATE against a baseline method, which uses one feature extracted without any technique, under a unified setting. For reference, we still

Table 1: The results of semantic correspondence (left, PCK@0.1) and label-scarce semantic segmentation (right, mIoU↑). **Red** for the best result and **blue** for the runner-up.

| Method | | $PCK_{img}$ ↑ | $PCK_{bbox}$ ↑ |
|---|---|---|---|
| Non-DF | DINO | 51.68 | 41.04 |
| | DHPF | 55.28 | 42.63 |
| DF | DIFT | - | 52.90 |
| | DHF | 72.56 | 64.61 |
| Baseline | nn | 61.15 | 51.66 |
| | conv | **73.96** | **65.74** |
| **GATE** | nn | 64.47 | 55.72 |
| | conv | **76.60** | **69.10** |

| Method | Bedroom-28 | Horse-21 |
|---|---|---|
| ALAE | 20.0 ± 1.0 | – |
| GAN Inversion | 13.9 ± 0.6 | 17.7 ± 0.4 |
| GAN Encoder | 22.4 ± 1.6 | 26.7 ± 0.7 |
| SwAV | 41.0 ± 2.3 | 51.7 ± 0.5 |
| SwAVw2 | 42.4 ± 1.7 | 54.0 ± 0.9 |
| MAE | 45.0 ± 2.0 | 63.4 ± 1.4 |
| DatasetGAN | 31.3 ± 2.7 | 45.4 ± 1.4 |
| DatasetDDPM | 47.9 ± 2.9 | 60.8 ± 1.0 |
| DDPM | **49.4 ± 1.9** | **65.0 ± 0.8** |
| **GATE** | **53.1 ± 2.7** | **67.2 ± 1.1** |

Table 2: Results on the two standard semantic segmentation datasets, ADE20K and CityScapes. **Red** for the best result and **blue** for the runner-up.

| Category | Method | ADE20K | | | CityScapes | | |
|---|---|---|---|---|---|---|---|
| | | mIoU↑ | aAcc↑ | mAcc↑ | mIoU↑ | aAcc↑ | mAcc↑ |
| SOTA | MaskCLIP | 23.70 | - | - | - | - | - |
| | ODISE | 29.90 | - | - | - | - | - |
| | VPD | **37.63** | **79.16** | **50.08** | **55.06** | **90.14** | **68.96** |
| Ours | GATE | **40.51** | **79.68** | **54.90** | **64.20** | **92.83** | **76.98** |

provide the results from four state-of-the-art methods: DINO [3] and DHPF [27] as non-DF methods, as well as DIFT [37] and DHF [24] as DF methods. The results are in the left part of Table 1.

**Semantic Segmentation.** The strong generalizability of a pre-trained diffusion model can ensure good discriminative performance even when labeled training data is scarce. For this scenario, we show the results in the right part of Table 1. The SOTA diffusion feature method for this setting, DDPM [1], serves as the major competitor. We also include other segmentation methods: DatasetGAN [59], DatasetDDPM, MAE [13], SwAV [2], GAN Inversion [40], GAN Encoder, VDVAE [4], and ALAE [32]. We further validate our method on the more common setting of semantic segmentation using standard datasets: ADE20K [61] and CityScapes [5], with the results presented in Table 2. The competitors are MaskCLIP [8], ODISE [50], and VPD [60], where VPD is re-evaluated with the diffusion model frozen for fairness.

Generally, GATE outperforms competitors by large margins on all datasets. It demonstrates that previous diffusion feature approaches have been hindered by content shift, and GATE has a promising application as a generic booster for feature quality.

## 6.3 Qualitative Analysis

In Figure 8, we provide feature visualization for qualitative analysis of GATE. The visualization is obtained using PCA analysis, reducing the channels of features to 3, which are regarded as RGB for visualization. We can observe: (i) The attention features become clearer and closer to the input image when more generation techniques are applied according to GATE, showing the suppression effect on content shift. (ii) Notably, for the second image where a person is riding a horse, the baseline attention feature fails to follow the instruction, *i.e.*, attending only to the horse and ignoring the person. In contrast, generation techniques applied according to GATE help attention features attend to the correct object. (iii) From convolutional features, we can see the application of generation techniques brings stronger diversity.

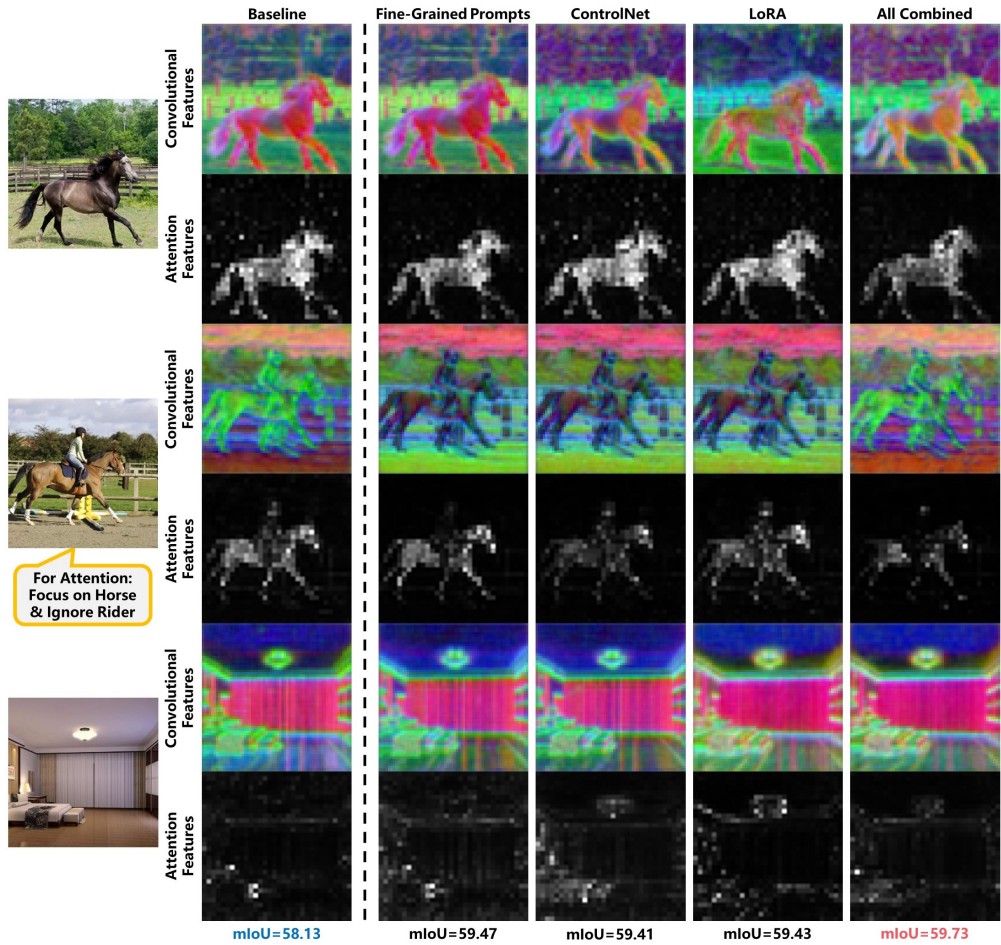

Figure 8: Effect of GATE without feature amalgamation. Images with various scenes are shown for generalizability. In the second image, the attention feature is asked to focus on the horse and ignore the rider. The mIoU performance is on a single Horse-21 split, with **red**/**blue** for the best/worst.

## 6.4 Ablation Study: Effect without Feature Amalgamation

For ablation study, we aim to evaluate the effect of selected techniques without feature amalgamation. The discriminative performance is shown at the bottom line of Figure 8, which is obtained on a single Horse-21 split instead of five random repeats for faster evaluation. We can observe: (i) Every individual technique can improve feature quality over baseline. (ii) When multiple techniques are applied simultaneously, stronger improvement can be obtained. This demonstrates that all three selected techniques can benefit feature quality, and their benefits can be combined together.

## 7 Conclusion and Future Work

In this paper, we reveal a phenomenon named content shift that has been causing degradation in diffusion features. Based on the analysis of its cause, we propose to suppress it with off-the-shelf generation techniques, which allows hitchhiking the advancements in generative diffusion models. This approach, while enjoying simplicity, is experimentally demonstrated to be generically effective.

However, the effectiveness of GATE relies on the selected techniques, for which we propose both a qualitative evaluation guideline and a quantitative metric. Though we selected three effective techniques and reported failed cases, there still is more to explore, which might potentially lead to more effective implementations. Furthermore, we only experimented with three tasks, so the full potential of GATE might remain under-explored.

## Acknowledgments

This work was supported in part by the National Key R&D Program of China under Grant 2018AAA0102000, in part by National Natural Science Foundation of China: 62236008, U21B2038, U23B2051, 61931008, 62122075, 62025604, 62206264, and 92370102, in part by Youth Innovation Promotion Association CAS, in part by the Strategic Priority Research Program of the Chinese Academy of Sciences, Grant No. XDB0680000, in part by the Innovation Funding of ICT, CAS under Grant No.E000000, in part by the China National Postdoctoral Program for Innovative Talents under Grant BX20240384.

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

# A Implementation Details and Efficiency Concerns

In this section, we will explain the implementation details of each selected technique and feature amalgamation. Further discussion will also be provided on how these techniques manage to suppress content shift and the efficiency impact of our implementation.

## A.1 Fine-Grained Prompts

Fine-grained prompts can be integrated into diffusion feature very simply. We only need to replace the simple and short prompts, such as "a photo of a horse" in most DF approaches [37, 9, 45, 55, 21], with more complex ones. Since prompts are a built-in function of current diffusion models, the only required integration of fine-grained prompts is to generate these prompts. To fulfill our goal, we need detailed descriptions of the image content, and many image captioning models can serve this purpose, such as Kosmos-2 [31]. It is also possible to manually design prompts that can cover various images, which is found to be almost as effective as auto captioners. When this approach is used, our observation is only in the training set to avoid data leaks. While this is effective for convolutional features, attention features require more consideration. To be specific, using specific prompts for each image causes inconsistency in the structure of attention features. To fill this gap, we always use manual and fixed prompt for attention features, even when convolutional features utilize auto-captioners. This fixed prompt is designed by observing images and describing the common objects. For example, "bedroom, a bed, some bedroom furniture, lights, a door, ceiling, floor, walls, pillow, quilt, chair and table, window, in good quality".

Regarding the efficiency of integrating fine-grained prompts, we do not make any changes to the original feature extraction pipeline but only introduce the overhead of image captioning. Theoretically, this is still linear complexity. In practice, the actual time consumption depends on what auto-captioner is used. Moreover, these prompts can be re-used for the extraction of many groups of features, further reducing the proportion of overhead in the total time.

## A.2 ControlNet

ControlNet requires a parameter, the control signal, which should be set as the input image itself to suppress content shift. More specifically, the control image ControlNet requires is not an ordinary image, but a specially processed one, such as a depth image or a canny image. Among all control image types, we find that canny images are exceptionally efficient because their process does not use any neural network model. Therefore, we use canny as the sole control type for feature extraction.

Although the simple usage is already yielding satisfying results, we find an additional way to further exploit ControlNet. It is reported that taking multiple denoising steps before feature extraction can make the attention feature less blurry [51]. However, taking multiple denoising steps is not a common practice for diffusion features, as it also brings severe performance drops caused by content shift. With ControlNet, it becomes possible to harness this good property without failing to suppress content shift. We only do this for attention features, so convolutional features are still extracted in a one-step manner.

Regarding efficiency, processing an image into a canny image takes a very small overhead, So the major overhead introduced by ControlNet should be extracting attention features with multiple denoising steps. Whilst this does take obviously longer time than one-step feature extraction, its impact is limited due to two factors: (i) We empirically find that the optimal effect comes with less than 10 denoising steps. If more steps are taken, attention features will meet unacceptable content shift even with ControlNet applied. (ii) Different from the diversity effect in convolutional features, attention features do not see obvious enhancement when amalgamated. Therefore, we only extract one attention feature per image and concatenate it to the amalgamation result of convolutional features. Considering the two factors, taking multiple denoising steps does not in fact contain many steps and its use is relatively rare in the overall feature extraction process. Thus, the efficiency impact of integrating ControlNet is small.

## A.3 LoRA

The integration of LoRA is also simple. Integrating LoRA into feature extraction only requires switching the base model weights to LoRA weights, so the major effort might be training a good LoRA weight. Fortunately, the community has offered many handy tools for ordinary users, such as kohya_ss[3], which can be directly utilized for our purpose. Following the tradition from the community, we choose 30 random images from the training set to train a LoRA weight. Specifically, we choose the *LoCon* type of LoRA, which additionally inserts trainable parameters between convolutional layers, and follow the corresponding guides for training.

The major overhead introduced by LoRA is its training. As a popular tool for AIGC players, the training for LoRA is feasible even on personal computers, showing its efficiency. In our experiments, training one LoRA weight usually takes less than 30 minutes. Moreover, the trained LoRA weights can be re-used, further reducing the proportion of overhead in the feature extraction process.

## A.4 Regularized Weight Assignment for Feature Amalgamation

Feature amalgamation can be accomplished by simply concatenating different features together. However, this is found suboptimal, leading to a superior strategy for feature amalgamation, which is weight assignment [24, 62]. Roughly, it assigns weights to each feature and amalgamates the features via a linear combination according to the weights. We next formalize this process. We have $b$ feature maps $\boldsymbol{r}_1, \boldsymbol{r}_2, \cdots, \boldsymbol{r}_b \in \mathbb{R}^{c \times w \times h}$ for each image, with $c, w, h$ being channel, width, and height, respectively. Weight assigners are simple MLP or CNN networks parameterized by $\theta$ and can be denoted as $f(\cdot; \theta)$. Considering that the downstream task might benefit from various perspectives, there are $n$ assigners. Assigning weights $w_i^j$ is denoted as:

$$w_i^j := f^j(\boldsymbol{r}_i; \theta^j) \in \mathbb{R}, \text{ s.t. } \sum_{i=1}^b w_i^j = 1, j = 1, \cdots, n \tag{3}$$

Afterward, we take the linear combinations of features and concatenate them as the final feature:

$$\boldsymbol{r} = \text{Concat}(\sum_{i=1}^b w_i^1 \boldsymbol{r}_i, \sum_{i=1}^b w_i^2 \boldsymbol{r}_i, \cdots, \sum_{i=1}^b w_i^n \boldsymbol{r}_i) \tag{4}$$

We notice that this weight assignment strategy is suboptimal. To be specific, the assigners tend to converge to a trivial solution, where they assign weights almost equally to all features and different assigners share a similar prediction. To tackle this issue, we introduce two extra regularization terms. The first one encourages sparsity, meaning the assigners should concentrate on only a few features. The second one, named diversity, promotes a larger discrepancy between the predictions of different weight assigners. Formally, the loss function is:

$$\mathfrak{L}_{fin} = \mathfrak{L} - \gamma_1 \sum_{j=1}^n \left\| \mathbf{w}^j \right\|_2 - \frac{\gamma_2}{2} \sum_{j=1}^n \sum_{k \neq j} \left\| \mathbf{w}^j - \mathbf{w}^k \right\|_2, \tag{5}$$

where $\mathfrak{L}$ is the original loss of the downstream task; $\gamma_1$ and $\gamma_2$ are the hyper-parameters of the regularization terms; $\mathbf{w}^j := (w_1^j, w_2^j, \cdots, w_b^j)^{\mathrm{T}}$, and $\| \cdot \|_2$ denotes the $\ell_2$-norm. Appendix D.5 will visualize the direct effect of the proposed regularization on weight distribution.

Since attention features do not show the same diversity effect as convolutional features, the two types of features are processed differently. To be specific, we obtain many convolutional features but only one attention feature per image. Feature amalgamation is conducted among convolutional features, while the attention feature is concatenated to the final amalgamation result.

Next, we aim to explain why the two proposed regularization terms can achieve their design purpose. Note that the weights of all features are restricted to sum up to 1, which means the $\ell_1$-norm of $\mathbf{w}$ is always 1. For a vector with a fixed $\ell_1$-norm, a larger $\ell_2$-norm means concentrating on less axes among all dimensions. Therefore, we use the negative value of its $\ell_2$-norm as sparsity regularization to promote a larger $\ell_2$-norm. As for diversity regularization, it simply measures and promotes the discrepancy among the predictions of different weight assigners.

---

[3]https://github.com/bmaltais/kohya_ss

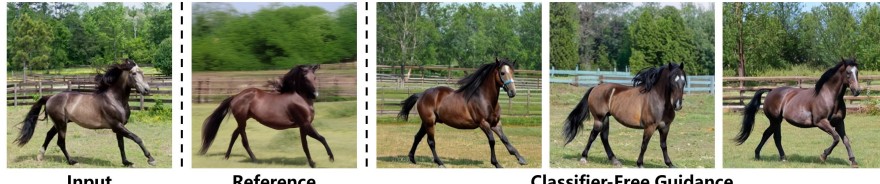

| Input | Reference | Classifier-Free Guidance |

Figure 9: Img2Img generation results of classifier-free guidance according to the GATE guideline, showing no obvious effect for suppressing content shift.

Table 3: Experimental results comparing the influence on suppressing content shift of ControlNet and IP-Adapter.

| Method | mIoU |
|---|---|
| DDPM | 65.0 ± 0.8 |
| GATE (Only IP-Adapter) | 65.8 ± 1.3 |
| GATE (Only ControlNet) | **66.2 ± 1.2** |
| GATE (Full) | **67.2 ± 1.1** |

Compared with previous weight assignment approaches [24, 62], we only add two regularization terms, thus introducing negligible overhead. Weight assignment itself does trade efficiency for performance. However, since this approach has already been adopted in previous approaches, we think it is rational to only consider the additional efficiency impact of our proposed regularization terms.

## B    Classifier-Free Guidance and IP-Adapter: Failed Case

Classifier-free guidance [16] is a widely adopted generation technique in current diffusion models. Its design purpose is to sacrifice diversity for better fidelity, similar to low-temperature sampling [18] in other generative models.

We first show its Img2Img generation results according to the GATE guideline in Figure 9. It is clear that it makes little difference to the similarity between outputs and the input, already suggesting that it lacks the potential for suppressing content shift. Since it can also be implemented simply, we integrate it into feature extraction as well and actually examine its effect, leading to the same conclusion as the previous quick evaluation following the GATE guideline.

Why is classifier-free guidance unable to suppress content shift? It is designed to gain fidelity at the cost of diversity. While better fidelity is irrelevant to content shift, it might be helpful to lower the diversity of reconstruction given the blank caused by noises. However, the less diverse reconstruction is still not guaranteed to be centered on the original content. Therefore, even though classifier-free guidance can reduce diversity, content shift will not be suppressed. We hope this failed case can provide more insights into how generation techniques affect content shift and how to select them.

IP-Adapter is a generation technique designed for image variation, which shares a similar architecture to ControlNet. By inputting images, IP-Adapter helps generate new images with some elements taken from the reference one. Since the goal of IP-Adapter is image variation instead of strict control, it is less effective in mitigating content shift than ControlNet. The weaker effectiveness of IP-Adapter is experimentally demonstrated on the Horse-21 dataset, as shown in Table 3. IP-Adapter is not entirely ineffective for content shift, but less effective than ControlNet and thus not included in our implementation.

## C    Experimental Details

### C.1    Implementation Details

For the diffusion model to extract features, we choose Stable Diffusion v1.5 [34] to be consistent with SOTA competitors. For semantic correspondence, two settings are examined: (i) directly performing

nearest neighbor algorithm [39] (*nn*), (ii) adding an extra convolutional layer after each assigner (*conv*). For label-scarce semantic segmentation, the downstream model follows the framework in [1], keeping hyper-parameters unchanged. For standard semantic segmentation, we instead use UPerNet [49] as the downstream model. $\gamma_1$ and $\gamma_2$, the hyper-parameters for the proposed regularization terms, are tuned for each dataset. Choosing the number of weight assigners is mainly a tradeoff between efficiency and performance (Appendix D.4). We use less than three based on our resources.

## C.2 Feature Extraction Details for Each Task

### C.2.1 Semantic Correspondence

Convolutional features include two features per setting under different random seeds:

(i) Two with fine-grained prompt "a photo of aeroplane, bicycle, bird, boat, bottle, bus, car, cat, chair, cow, dog, horse, motorbike, person, potted plant, sheep, train, tv monitor, high quality, best quality, highly realistic, masterpiece, high resolution".

(ii) Two with LoRA.

(iii) Two with ControlNet.

Attention features are obtained using a prompt including all object categories, with ControlNet (denoising from $t = 60$) and LoRA also applied.

### C.2.2 Semantic Segmentation on Horse-21

Convolutional features include:

(i) A feature using fine-grained prompt "a horse, high quality, best quality, highly realistic, masterpiece".

(ii) A feature using ControlNet, with a simple prompt "a horse".

(iii) Two features using different LoRA weights and a simple prompt. The first LoRA weight is trained to generate high-quality images, while the second LoRA weight is trained until it slightly overfits.

Attention features are obtained using a prompt "a photo of a single horse running in a sports field, with a well-equipped rider on its back, seems they are in a competition, high quality, best quality, highly realistic, masterpiece", with ControlNet and the first LoRA weight applied. Additionally, we concatenate the output feature of amalgamation with a feature extracted using DDPM [1].

### C.2.3 Semantic Segmentation on Bedroom-28

Convolutional features include:

(i) A feature using fine-grained prompt "a photo of a tidy and well-designed bedroom". It is found that quality prompts such as "high quality" will be interpreted as the quality of the room instead of the image, so such prompts are not utilized.

(ii) A feature using ControlNet, with a simple prompt "a bedroom".

(iii) One feature using LoRA weight, which is moderately trained to generate high-quality images.

Attention features are obtained using a prompt "bedroom, a bed, some bedroom furniture, lights, a door, ceiling, floor, walls, pillow, quilt, chair and table, window, in good quality". ControlNet and LoRA are also applied. Additionally, we concatenate the output feature of amalgamation with a feature extracted using DDPM [1].

### C.2.4 Semantic Segmentation on ADE20K

Convolutional features include:

(i) One feature using only fine-grained prompt "a highly realistic photo of the real world. It can be an indoor scene, or an outdoor scene, or a photo of nature. high quality" plus all category labels.

Table 4: Examining GATE on SDXL features. The best result is marked as red. For *Baseline*, three basic feature maps are used. For other settings, one or two basic feature maps are replaced with technique-enhanced ones, keeping the total number of features unchanged.

| Category | Basic | Prompt | ControlNet | mIoU↑ |
|----------|-------|--------|------------|-------|
| Baseline | ✓ | | | 53.40 |
| Individual | ✓ | ✓ | | 53.61 |
| | ✓ | | ✓ | 53.66 |
| Combined | ✓ | ✓ | ✓ | **53.96** |

(ii) Two features using the same prompt as above and additionally ControlNet and LoRA. The two features use two different LoRA weights, where one is moderately trained and the other slightly overfits.

No attention features are used. We find that in very complex scenes attention features can be of low quality and bring slight degradation instead of enhancement.

### C.2.5 Semantic Segmentation on CityScapes

Convolutional features include:

 (i) One feature using only fine-grained prompt "An urban street scene with multiple lanes, various buildings, traffic lights, cars in the lanes, and pedestrians, highly realistic".
(ii) Two features using the same prompt as above and additionally ControlNet and LoRA. The two features use two different LoRA weights, where one is moderately trained and the other slightly overfits.

No attention features are used, based on the same observation as for ADE20K.

### C.3 Details for Exploration of Content Shift

For CIFAR10, high-quality and low-quality prompts are as follows:

> "cinematic shot photo taken by ARRI, photo taken by sony, incredibly detailed, sharpen, masterpiece, best quality, realistic, HD, raytracing, CG unified 8K wallpapers"
> "low quality, low resolution, blurry, draft, grainy, disfigured, deformed, low contrast, underexposed, overexposed, bad art"

For semantic segmentation, they are:

> "a horse, high quality, best quality, highly realistic, masterpiece"
> "a horse, low quality, worst quality, deformation, broken shape, blurry"

## D    Additional Experiments

### D.1    Additional Qualitative Results

Besides the feature visualization in Figure 8, we provide additional visualization here in Figure 10.

### D.2    Demonstration for Generalizability

We would like to further demonstrate the generalizability of GATE by applying it to a different diffusion model, SDXL [33]. The architecture of SDXL has been significantly modified to include more complicated attention structures while its overall upsampling blocks are simplified. Due to the architecture change, the previous method for feature extraction cannot fully exploit the potential of SDXL, thus making the performance lower than Stable Diffusion v1.5. Nevertheless, it is enough to

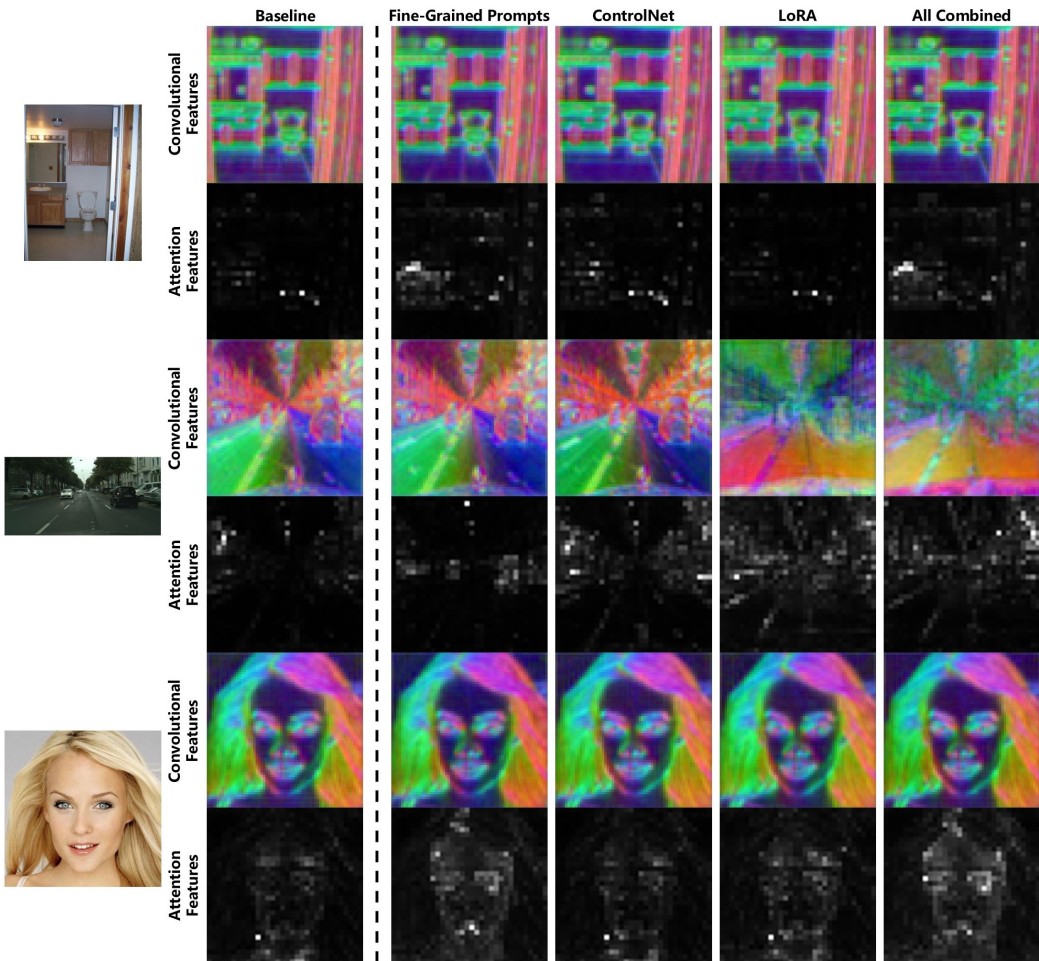

Figure 10: Additional feature visualization for qualitative analysis with different types of images, showing the effectiveness of GATE on both convolutional features and attention features.

Table 5: Results on Horse-21 to study the effect of feature amalgamation, using the same setting as Figure 8. Each column corresponds to a group of features. *Basic** is extracted at a different timestep with other settings remaining the same as *Basic*.

| Basic | Basic* | Prompts | ControlNet | LoRA | mIoU |
|-------|--------|---------|------------|------|------|
| ✓ | | | | | 58.13 |
| ✓ | ✓ | | | | 59.44 |
| ✓ | | ✓ | | | 60.12 |
| ✓ | | | ✓ | | 60.02 |
| ✓ | | | | ✓ | 60.27 |
| ✓ | | ✓ | ✓ | ✓ | **60.74** |

demonstrate the generalizability of GATE. Furthermore, since LoRA training for SDXL is different and requires more investigation, we have excluded this technique from this small experiment.

Observing the results in Table 4, we can again confirm the benefit of GATE. Both techniques can enhance performance. Moreover, combining two techniques is able to bring further improvement. This demonstrates the generalizability of GATE as it can be integrated into different diffusion models.

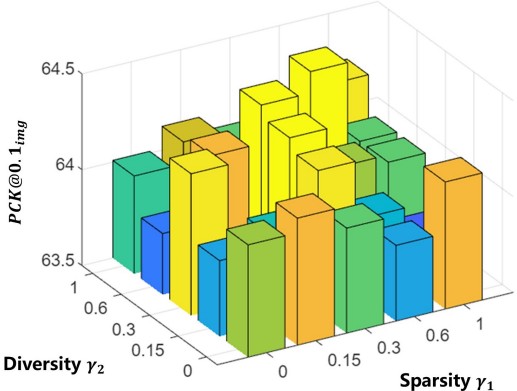

Figure 11: Sensitivity analysis of the regularization terms, where $\gamma_1/\gamma_2$ is for sparsity/diversity. These results validate the necessity of these regularization terms.

### D.3 Effect of Feature Amalgamation

Now we aim to examine the effect of our regularized weight assignment strategy for feature amalgamation. We will first demonstrate that amalgamating many features leads to better performance than a single feature. Then we will conduct an analysis of the hyper-parameter sensitivity regarding the proposed regularization terms.

The first experiment is conducted on the same Horse-21 split as Figure 8. In addition, this experiment only applies techniques individually, not simultaneously, for a simpler setting. The following observations can be made from Table 5:

 (i) Amalgamating many features is generally superior to a single feature. Even if we simply use features obtained from different timesteps, it is beneficial.

 (ii) If the features are from different technique combinations rather than simply different timesteps, the performance will be higher. This shows that applying various techniques can yield more diverse diffusion features.

(iii) If more features are used, the performance will be better than when only two are used.

To analyze the sensitivity property of the two regularization terms, we perform a grid search on the semantic correspondence task. The reported metric is PCK@0.1$_{\mathrm{img}}(\uparrow)$ and the model type is *nn*. Based on Figure 11, we can conclude that both regularization terms contribute to the performance gain, but they need tuning. We have tuned them for the results in Table 1.

### D.4 Ablation Study on Number of Weight Assigners

In this part, we provide a supplementary ablation study on the number of weight assigners. The results in Table 6 show that when we increase the number of weight assigners, the overall performance will improve gradually. However, we have two following concerns:

 (i) More weight assigners will take a longer time to converge. The training steps that are enough for fewer assigners to fully converge are insufficient for more assigners, indicated by that the the performance of more assigners still improves with more training steps.

 (ii) Increasing the number of weight assigners will bring more memory usage and more time consumption, which will affect efficiency.

Consequently, choosing the number of weight assigners is mainly a tradeoff between performance and efficiency.

Table 6: Results (mIoU↑) to reveal the impact of adjusting the number of weight assigners. The best result is marked as red.

| Num | $1 \times$ Training Steps | $2 \times$ Training Steps | Gain |
|---|---|---|---|
| 1 | 64.58 | 63.91 | -0.67 |
| 2 | 65.39 | 65.68 | 0.29 |
| 3 | 65.44 | **65.84** | 0.40 |
| 4 | 65.19 | 65.73 | 0.54 |

Table 7: Experimental results on image classification in comparison to both ResNet and a diffusion feature baseline without GATE.

| Method | Acc(%) |
|---|---|
| ResNet-50 | 93.62 |
| Baseline | **94.55** |
| GATE | **95.21** |

### D.5 Impact of Regularization on Weights

We aim to visualize the impact of our proposed regularization terms on weight distribution, to show that their effect aligns with the design purpose. The statistics are shown in Figure 12. We select the semantic correspondence task under *nn* setting with two weight assigners.

The first four figures visualize the impact of sparsity. The first two show the Gaussian probability density fitted to the mean and variance of top-1 and top-2 weights. The other two show the average value of all weights. Without regularization, top-1 weights will be rather small. With regularization, top-1 weights will become much larger, showing that the prediction of assigners becomes more concentrated.

The last two figures look into diversity. We compare the predictions of two assigners. If no overlap is observed in top-1 nor top-2 weights, it counts as *no overlap*, which is the best case. If overlap occurs in top-1 weights, it is regarded as *overlap @ top-1*. If overlap only occurs when we consider both top-1 and top-2 weights, this is *overlap @ top-2*. It is clear that diversity regularization can effectively force assigners to focus on different feature maps, thus providing information from various perspectives.

### D.6 Image Classification

Although image-level tasks are not typically utilized for the evaluation of diffusion features, we have used this task for early empirical studies. Hence, in Table 7, we provide some results on CIFAR10, in comparison to a standard ResNet backbone[4] and a baseline where diffusion features are used without GATE.

## E Future Direction

Long tail and out-of-distribution scenarios are challenging problems for discrimination, where a model that works well under i.i.d. settings can degrade significantly [43]. There have been a few attempts to apply diffusion models to such challenging scenarios [10, 46, 56], but more efforts are still considered helpful. Recently, there has been a study [11] that might open more opportunities to this topic. To be specific, AUC is a useful metric and loss function for long tail and out-of-distribution studies [41, 42], but it used to be not applicable to pixel-level tasks such as semantic segmentation. This recent study, however, has successfully introduced a new AUC-based tool that can be applied to segmentation as well, enabling more future studies in this direction.

---

[4] https://github.com/kuangliu/pytorch-cifar

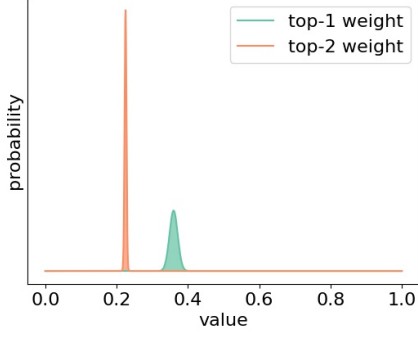

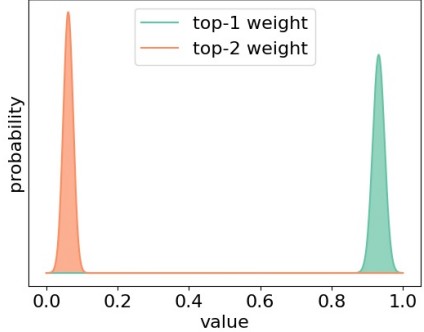

(a) Probability density of weights **w/o** regularization.

(b) Probability density of weights **w/** regularization.

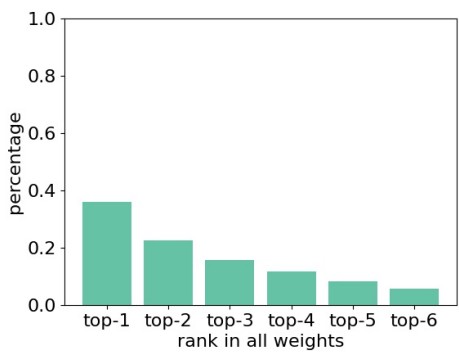

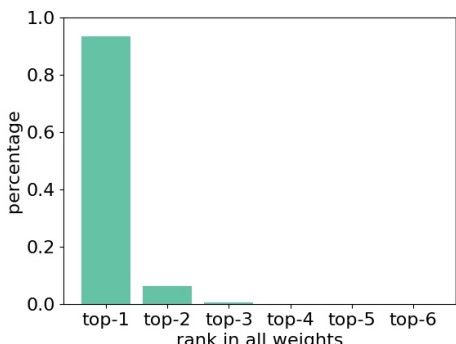

(c) Average values of all weights **w/o** regularization.

(d) Average values of all weights **w/** regularization.

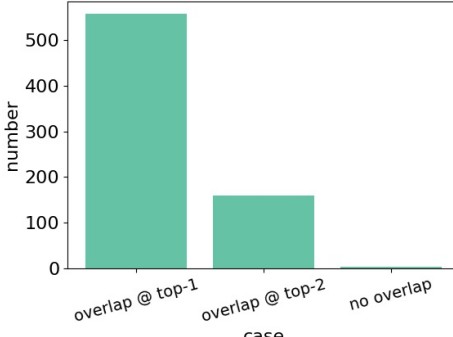

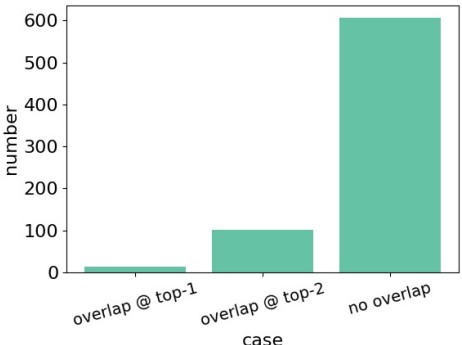

(e) Overlap issues of top-2 weights **w/o** regularization.

(f) Overlap issues of top-2 weights **w/** regularization.

Figure 12: Visualization of weight distribution with and without regularization. The first four focus on sparsity and the other two visualize the impact of diversity.

## F  Computation Resources

We use Nvidia(R) RTX 3090 and Nvidia(R) RTX 4090 GPUs for the experiments, all with 24GB VRAM.

The experiments on semantic correspondence take about 5 hours per run and about 50GB of disk storage. The experiments on Horse-21 and Bedroom-28 take about 5 hours per run, where one run consists of 5 repeats on different splits. Furthermore, each run uses about 45GB of disk storage. The experiments on ADE20K and CityScapes take about 2 days per run with no additional disk usage besides what is needed to store model checkpoints and datasets. Our early exploration of content shift utilizes relatively lightweight experiments for efficiency.

## G  Asset License

- SPair-71k: Available at https://cvlab.postech.ac.kr/research/SPair-71k/.
- Horse-21 and Bedroom-28 (LSUN): Available at https://github.com/fyu/lsun.
- ADE20K: Custom (research-only, non-commercial), at https://groups.csail.mit.edu/vision/datasets/ADE20K/terms/.
- CityScapes: Custom, at https://www.cityscapes-dataset.com/license/.

