# OpenReview forum: "Suppress Content Shift: Better Diffusion Features via Off-the-Shelf Generation Techniques"
_NeurIPS.cc/2024/Conference — NeurIPS 2024 poster_

### Official Review · Reviewer_RvKH · 2024-07-02

**Soundness:** 2
**Presentation:** 1
**Contribution:** 2
**Rating:** 4
**Confidence:** 3

**Summary:**

The paper describes a pipeline for improving feature extraction from pre-trained diffusion models for general-purpose task such as semantic segmentation. The focus is on "content shift", where extracted feature maps at a certain time step in the diffusion process show differences in the content composition compared to the input image. Three previous methods for controlling image to image transformation with diffusion models are used for extracting features and attention maps. Subsequently, these are fused by a learned weighted linear combination, or amalgamation, providing updated image features for use in the downstream application. The pipeline is tested for semantic correspondence on the SPair-71k dataset, and for semantic segmentation on the Bedroom-28, Horse-21, ADE20K, and CityScapes datasets. The results show improvement over the compared baselines (vanilla feature extraction, and some other previous feature extraction techniques).

**Strengths:**

+ The focus on improving features from diffusion models for downstream applications is interesting, and to my knowledge a direction with relatively little previous work
+ The evaluation shows favorable results, where the proposed pipeline can improve on vanilla features and some other feature extraction methods

**Weaknesses:**

- The paper is written on a quite high level, with many of the relevant details left out or put in the appendix
- The technical contribution is rather limited, combining previous techniques for the purpose of improving diffusion features
- The evaluation is lacking, with many details missing and without proper ablation study

**Questions:**

* In my view, there are too many details left out in the main paper, related to, e.g., method, datasets, and experiments. In its current state, it relies too much on details in the appendix. For example, the dataset used for semantic correspondence in Tab. 1 is not even specified, metrics are not explained, implementation details are missing, etc. To me, this is more important compared to the lengthy and speculative motivation of content shift in Section 4. It is rather obvious that features can shift away from the input image with excessive noise added. I would suggest moving most of these discussions and figures to the appendix, and instead provide the necessary details for understanding the pipeline and experiments.

* Although the appendix provides further information, there are still missing details and experiments. For example, one central hyper-parameter is the noise level used for feature extraction. This is neither specified nor tested in experiments. What noise levels were used? How were these selected? How is the improved performance of the suggested pipeline reflected over different selections of noise levels? Is it possible to combine a larger number of different noise levels, e.g., in the learned combination?

* Rather than the speculative motivation in Section 4, is there a more systematic way of quantifying the content shift? As it looks now, this is rather speculative and only verified through a few example images. Since this is the main focus, it would be important to measure and compare, e.g., how it behaves over different noise levels and for different feature extraction techniques.

* The experimental setup now relies on having rather specific types of images (horses, bedrooms, street view, etc.), so that a general text prompt can be provided for the fine-grained prompt image generation. How can this generalize to more diverse types of images?

* In Fig. 5, there are some examples for classification on CIFAR-10, but there are no experimental results on classification. How does the proposed method perform in classification tasks? Isn't this the most straightforward strategy for evaluation of feature quality?

* How are clean features obtained from the noisy features, in Fig. 4?

* In summary, although there are interesting observations, the manuscript is to me not yet mature enough, and lacking both in details and experiments.

**Limitations:**

* Limitations are only very briefly mentioned in the conclusions. These could be elaborated on, in connection to the points discussed above.

---

> ### Author Rebuttal · Authors · 2024-08-06
>
> Thanks for constructive comments. We would like to response as follows.
>
> >**Q1:**
> The paper is written on a quite high level with many details left out or in appendix.
>
> **A1:**
> We focused on content shift and how generation techniques affect it. Some details were thus omitted in the main body due to the space limitation, which might be confusing. We will revise according to your advice.
>
> For pipeline, please refer to **global question GQ1**.
> For experimental details, we show **part of** refinement below due to space limit.
>
> We extract features at $t=50$.
> - Semantic Segmentation
> 	- Convolutional features:
> 		- One with fine-grained prompt "a highly realistic photo of the real world. It can be an indoor scene, or an outdoor scene, or a photo of nature. high quality" plus all category labels (ADE20K).
> 		- Two features: same prompt as above, ControlNet and two different LoRA weights.
> 	- No attention features are used, as they do not bring benefit in complex scenes.
>
> >**Q2:**
> The technical contribution is limited, combining previous techniques.
>
> **A2:**
> Perhaps due to presentation, it is a pity to leave you this impression.
> In fact, our main contribution is the pioneering discovery of content shift, a widespread, harmful, yet hidden behavior of diffusion features.
> We systematically investigated its cause and thus found that existing generation techniques are enough to mitigate such shift effectively.
> Hence, we choose these techniques for easier implementation and extension.
> The key point is: among numerous techniques, which can serve for discrimination tasks?
> Our GATE guideline bridges this gap and inspires the necessary adaptation to these techniques.
> We hope all these observation, analysis, guidance, and adaptation can benefit future work.
>
>
> >**Q3:**
> More experimental details and ablation study.
>
> **A3:**
> For more evaluation and details, please refer to **GQ4** and your **Q1**.
> Fig.10 serves as ablation. For your convenience, we list results below, which validate the effectiveness of each technique and their combination.
>
> |Method|mIoU (on a single split)|
> |:-|:-:|
> |Baseline|58.13|
> |GATE (Fine-grained prompts)|59.47|
> |GATE (ControlNet)|59.41|
> |GATE (LoRA)|59.43|
> |GATE (Full)|**59.73**|
>
> >**Q4:**
> Provide necessary details for pipeline and experiments in the main content.
>
> **A4:**
> We will refine manuscript according to your suggestions.
> The pipeline in **GQ1** will replace Fig.2 in Preliminary.
> The experimental details in **Q1** will be added.
>
> >**Q5:**
> What noise levels were used? Is it possible to combine different noise levels?
>
> **A5:**
> In fact, the choice of noise level is orthogonal to our work.
> Early diffusion feature work [2] systematically analyzed this issue and suggested $t=50$ as a general choice. We observe similar results and thus follow this practice.  As shown below, we test ControlNet with other noise levels on a single Horse-21 split, where GATE outperforms baseline across noise levels, especially when the noise level is stronger.
>
> |Method|t=400|t=700|
> |:-|-|-|
> |Baseline|47.09|24.39|
> |GATE|**56.35**|**52.65**|
>
> It is reasonable to combine different noise levels, which is actually the focus of some prior arts [23, 56], including a semantic correspondence SOTA.
> Nevertheless, The results in Tab.1 show that amalgamating different generation techniques brings stronger enhancement.
>
> >**Q6:**
> A more systematic way of quantifying content shift.
>
> **A6:**
> We also developed a quantitative metric.
> We set feature extracted at $t=0$ as reference $FEAT_{ref} \in \mathbb{R}^{c\times h\times w}$ as it is less affected by noises. We use Laplacian operator to evaluate the contour difference between feature $FEAT$ and the reference:
>
> $$
> diff=\sum_{i,j}^{h\times w}|\left\|Laplacian(FEAT_{ref},i,j)\right\|_2-\left\|Laplacian(FEAT,i,j)\right\|_2|
> $$
>
> We then set a feature with stronger content shift as anchor and compare $diff_{anchor}$ with other features.
>
> $$
> Score=\frac{(diff_{anchor}-diff)}{diff_{anchor}}\in(-\infty,1]
> $$
>
> $Score=1$ means exact match, and a smaller value indicates more shift. Below are quantitative results obtained on a single Horse-21 split, showing $Score$ is consistent with mIoU performance.
>
> |mIoU|$Score$|
> |-|-|
> |59.3|0.73|
> |58.7|0.73|
> |54.2|0.20|
> |50.1|-0.17|
>
> In our manuscript, we use qualitative GATE guideline because it can well approximate $Score$ and is much simpler. According to your advice, we will add this metric in refinement.
>
>
> >**Q7:**
> How can the method generalize to more diverse types of images?
>
> **A7:**
> Among the three techniques, ControlNet and LoRA are insensitive. For fine-grained prompt, generalization is clarified in **GQ3**.
>
> >**Q8:**
> How does the method perform in classification tasks?
>
> **A8:**
> Diffusion features have much more channels (3520) than input images (RGB). Hence, prior arts prefer pixel-level task to image-level task like image classification.
> In Sec.4, we aim to demonstrate quality described by prompts should resemble image quality, so the classification dataset CIFAR10 is introduced for low-quality images.
>
> According to your advice, we provide additional results on classification in **GQ4**.
>
> >**Q9:**
> How are clean features obtained from noisy features?
>
> **A9:**
> - Diffusion models learn from pre-training to reconstruct clean representations from noisy inputs. Notably, this is also related to content shift as noisy inputs cause information loss.
> - Shortcut structures in UNet pass high-frequency details to upsampling stage, helping reconstruction.
>
> > **Q10:**
> Limitations are too brief.
>
> **A10:**
> We focus on content shift and its solution. Due to space limit, we omit many implementation details, which will be revised.
> Besides, effectiveness of GATE relies on selected techniques, for which we propose qualitative evaluation guideline. Though we select three effective techniques and report fail cases, there still is more to explore.
> Furthermore, we only experiment on three tasks, so the full potential of GATE might remain under-explored.

---

> > ### Comment · Reviewer_RvKH · 2024-08-13
> >
> > Thanks for the exhaustive reply and clarifications! I agree that the analysis and method are of good value, and the clarifications made in the rebuttal addresses many of the problems with limited information. I will revise my score accordingly. However, my worry is that the large amount of additional information provided through the rebuttal is too much to add in a revision.

---

> > > ### Author Response · Authors · 2024-08-13
> > >
> > > We greatly appreciate your diligence, expertise, and efforts! Through the responses, we pursue better ways to clarify some important concepts of our method, while the core idea and main contributions stay unchanged. Thanks to the discussion with all reviewers, we have polished our presentation and integrated the responses. It is a pity that according to the policy, we cannot provide the revised PDF directly. To dispel your concern about the amount of such revision, we would like to clarify how each response will be included in the revised manuscript.
> > >
> > > Briefly, although with additional information, most responses require no additional space or only need tolerable space to complete the revision (**GQ1** - **GQ4**, **Q2** - **Q5**, and **Q7** - **Q10**). Although **Q1** and **Q6** do require additional space, the policy has considered such revision and allows one additional page in the camera-ready version. Hence, if this paper is fortunately accepted, we can easily add all these responses in a revision. Even if we do not have the additional page, the revision is also feasible by following your advice to re-organize Fig.3 - Fig.6.
> > >
> > > **GQ1**: In this response, we add more details about the pipeline, especially how these features are extracted. In the revised version, these details will replace part of the previous preliminary section (Sec.3). Meanwhile, the new figure about pipeline details will replace Fig.2 in the preliminary section. All the revisions will take up little additional space.
> > >
> > > **GQ2**: We analyze two failure cases in this response, one of which has already been included in Appendix. For revision, we will put the other one in Appendix as well, not taking up the space of the main body.
> > >
> > > **GQ3**: This response explains how the three selected techniques generalize to images of various types, which was not clearly illustrated before. According to your advice, the experiment section (Sec.6) will include some necessary implementation details on the ADE20K dataset to clarify this issue.
> > >
> > > **GQ4**: The experimental results in this response will be organized as follows.
> > > - We add more metrics for the experiments on ADE20K and CityScapes. These new results will be directly integrated into the blank space in Tab.2, requiring no additional vertical space.
> > > - We provide the classification performance on CIFAR10. These results will be put in Appendix, since it is not a common practice to conduct experiments on this task.
> > >
> > > **Q1**: We provide many experimental details in this response. In the revision, according to your advice, we will summarize some important details such as datasets, metrics, and implementation details. The full version, which has been presented in the global comment, will be put in the Appendix due to space limitation.
> > >
> > > **Q2**: We will revise the introduction to better clarify the focus and novelty of this study.
> > >
> > > **Q3**: This response clarifies the ablation study that has been presented in Fig.10. Perhaps due to our presentation, the reader might ignore the ablation results presented at the bottom of Fig.10. For better clarity, we will split the discussion on ablation study and qualitative analysis, which only requires simple re-organization.
> > >
> > > **Q4**: please refer to the revision of **GQ1** and **Q1**.
> > >
> > > **Q5**: We will add necessary experimental details to clarify the noise level we use.
> > >
> > > **Q6**: In this response, we add the quantitative metric to measure content shift. We will add this content in Sec.5, requiring some additional space.
> > >
> > > **Q7**: please refer to the revision of **GQ3**.
> > >
> > > **Q8**: please refer to the revision of **GQ4**.
> > >
> > > **Q9**: This response clarifies how clean features are obtained from noisy features. We will revise the discussion in Sec.4. to clarify this issue. According to your advice, we can reduce some empirical results if it is necessary.
> > >
> > > **Q10**: We will revise the conclusion section.
> > >
> > > ---
> > >
> > > For your convenience, we attach the revised content right here.

---

> > > ### Author Response · Authors · 2024-08-13
> > >
> > > (Part 3 of 4)
> > > ## 6.1 Experimental Settings
> > >
> > > **Task \& Dataset.**
> > > Typically, diffusion feature studies \cite{baranchuk2021label, xu2023open, zhao2023unleashing} prefer fine-grained pixel-level tasks for evaluation.
> > > Following this practice, we select three tasks for experiments: semantic correspondence using SPair-71k \cite{min2019spair} dataset, label-scarce semantic segmentation using Bedroom-28 \cite{yu2015lsun} and Horse-21 \cite{yu2015lsun} datasets, and standard semantic segmentation using ADE20K \cite{zhou2017scene} and CityScapes \cite{Cordts16city} datasets.
> > >
> > > **Evaluation Metrics.**
> > > (i) For semantic correspondence, $\text{PCK\@0.1}\_{\text{img}}(\uparrow)$ and $\text{PCK\@0.1}\_{\text{bbox}}(\uparrow)$ are used, following the widely-adopted protocol reported in \cite{min2019spair}. (We omit @0.1 to save some space in Table 1.) These two metrics mean the percentage of correctly predicted keypoints, where a predicted keypoint is considered to be correct if it lies within the neighborhood of the corresponding annotation with a radius of $0.1 \times max(h, w)$. For $\text{PCK\@0.1}\_{\text{img}}$/$\text{PCK\@0.1}\_{\text{bbox}}$, $h, w$ denote the dimension of the entire image/object bounding box, respectively.
> > > (ii) For semantic segmentation, we use mIoU metric, which is the mean over the IoU performance across all semantic classes \cite{hao2020brief}. For each image, IoU (Intersection over Union, $\uparrow$) is defined by \#(overlapped pixels between the prediction and the ground truth) / \#(union pixels of them).
> > > In addition, we also use aAcc and mAcc, where aAcc is the classification accuracy of all pixels and mAcc averages the accuracy over categories.
> > >
> > > **Implementation Details.**
> > > For all tasks, we extract features at $t=50$, which is an optimal noise level suggested in \cite{baranchuk2021label}.
> > > When ControlNet is applied, except for standard semantic segmentation, we additionally start multi-step denoising from $t=60$.
> > > For feature amalgamation, we extract multiple convolutional features and one attention feature per image:
> > > - Semantic correspondence: We obtain six in total convolutional features using individual fine-grained prompt, ControlNet, and LoRA techniques, and one attention feature using a prompt including all object categories, with ControlNet and LoRA.
> > > - Label-scarce semantic segmentation: We obtain one convolutional feature using fine-grained prompts, one convolutional feature using ControlNet, and one (Bedroom-28) to two (Horse-21) features using different LoRA weights. One attention feature is extracted with all three techniques applied.
> > > - Standard semantic segmentation: One convolutional feature is obtained using only fine-grained prompts and two more are extracted additionally with ControlNet and different LoRA weights. One attention feature is extracted with all three techniques applied.
> > >
> > > Notably, the ADE20K dataset for standard semantic segmentation contains images of varying scenes, which can test how well the three techniques can generalize in this scenario.
> > > While ControlNet and LoRA are essentially insensitive to this scenario, very specific fine-grained prompts can meet some problems.
> > > To this end, we use a prompt that can cover different scenarios: "a highly realistic photo of the real world. It can be an indoor scene, or an outdoor scene, or a photo of nature. high quality".
> > > This prompt both covers various scenes for generalizability and describes the intended image quality for fine-grained effect.
> > >
> > > ## 6.2 Comparison with SOTA
> > >
> > > The experimental results are shown in Table 1 and Table 2.
> > > For most SOTA competitors, we borrow the reported results from their original studies.
> > > However, MaskCLIP and ODISE only provide results on ADE20K and it is hard to extend their implementations to CityScapes, so their results on CityScapes are missing.
> > > Furthermore, the original results reported by VPD are based on full-scale fine-tuning of diffusion UNet, which is not fair as we do not train the diffusion model.
> > > Therefore, we re-evaluated VPD with the diffusion UNet frozen and reported our results.
> > >
> > > (Remaining parts of the original subsection)

---

> > > ### Author Response · Authors · 2024-08-13
> > >
> > > (Part 4 of 4)
> > > ## 6.3 Qualitative Analysis
> > >
> > > In Figure 10, we provide feature visualization for qualitative analysis of GATE.
> > > The visualization is obtained using PCA analysis, reducing the channels of features to 3, which are regarded as RGB for visualization.
> > > We can observe:
> > > (i) The attention features become clearer and closer to the input image when more generation techniques are applied according to GATE, showing the suppression effect on content shift.
> > > (ii) Notably, for the second image where a person is riding a horse, the baseline attention feature fails to follow the instruction, *i.e.*, attending only to the horse and ignoring the person.
> > > In contrast, generation techniques applied according to GATE help attention features attend to the correct object.
> > > (iii) From convolutional features, we can see the application of generation techniques brings stronger diversity.
> > >
> > > ## 6.4 Ablation Study: Effect without Feature Amalgamation
> > >
> > > For ablation study, we aim to evaluate the effect of selected techniques without feature amalgamation.
> > > The discriminative performance is shown at the bottom line of Figure 10, which is obtained on a single Horse-21 split instead of five random repeats for faster evaluation.
> > > We can observe:
> > > (i) Every individual technique can improve feature quality compared to baseline.
> > > (ii) When multiple techniques are applied simultaneously, stronger improvement than each individual technique can be obtained.
> > > This demonstrates that all three selected techniques can benefit feature quality, and their benefits can be combined together.
> > >
> > > ---
> > >
> > > # 7 Conclusion and Future Work
> > >
> > > In this paper, we reveal a phenomenon named content shift that has been causing degradation in diffusion features.
> > > Based on the analysis of its cause, we propose to suppress it with off-the-shelf generation techniques, which allows hitchhiking the advancements in generative diffusion models.
> > > This approach, while enjoying simplicity, is experimentally demonstrated to be generically effective.
> > >
> > > However, the effectiveness of GATE relies on the selected techniques, for which we propose both a qualitative evaluation guideline and a quantitative metric.
> > > Though we selected three effective techniques and reported failed cases, there still is more to explore, which might potentially lead to more effective implementations.
> > > Furthermore, we only experiment on three tasks, so the full potential of GATE might remain under-explore
> > >
> > > ---
> > >
> > > >In Appendix
> > >
> > > # C IP-Adapter: Failed Case
> > > IP-Adapter is a generation technique designed for image variation, which shares a similar architecture to ControlNet.
> > > By inputting images, IP-Adapter helps generate new images with some elements taken from the reference one.
> > > Since the goal of IP-Adapter is image variation instead of strict control, it is less effective in mitigating content shift than ControlNet.
> > > The weaker effectiveness of IP-Adapter is experimentally demonstrated on the Horse-21 dataset, as shown in Table 3.
> > >
> > > |Method|mIoU|
> > > |:--|:-:|
> > > |DDPM|65.0 ± 0.8|
> > > |GATE (Only IP-Adapter)|65.8 ± 1.3|
> > > |GATE (Only ControlNet)|66.2 ± 1.2|
> > > |GATE (Full)|**67.2 ± 1.1**|
> > >
> > > ---
> > >
> > > >In Appendix
> > >
> > > ## E.5 Image Classification
> > >
> > > Although image-level tasks are not typically utilized for the evaluation of diffusion features, we have used this task for early empirical studies.
> > > Here, we provide some results on CIFAR10, in comparison to a standard ResNet backbone and a baseline where diffusion features are used without GATE.
> > >
> > > |Method|Acc(\%)|
> > > |:-|:-:|
> > > |ResNet-50|93.62|
> > > |Baseline|94.55|
> > > |GATE|95.21|

---

> ### Author Response · Authors · 2024-08-13
>
> (Part 1 of 4)
> >In Introduction line 41-51
>
> In pursuit of a method to suppress content shift, we need to further investigate why this phenomenon exists.
> We notice in Section 4 that the diffusion backbone reconstructs clean inner representations from noisy inputs in the middle of UNet before predicting noises based on the reconstructed content.
> The diffusion features we are using are in fact the reconstructed representations, which answers why we can obtain clean features from noisy images.
> However, since high-frequency details are potentially blurred out by noises in the inputs and recovered by "imagination" gained from vast pre-training, this reconstruction process inherently suffers from the risk of drifting from the original image.
> Content shift in diffusion features, naturally, reflects the drift during reconstruction.
> Consequently, to suppress content shift, we need an additional way to directly send the original clean image into the middle of UNet and steer the reconstruction towards the original image.
> To our delight, we notice that many off-the-shelf image generation techniques for diffusion models \cite{zhang2023adding, mou2023t2i, ye2023ip} also work by sending additional information into UNet and thus steering the reconstruction.
> Hence, it is possible to select some techniques that can directly satisfy the goal of suppressing content shift, which eases the implementation and extension of our method.
> Although the huge number of generation techniques might pose difficulty to the selection, we propose a guideline in Section 5 to bridge the gap.
> This method is denoted as *GenerAtion Techniques Enhanced* (**GATE**) diffusion feature and its effect is also shown in Figure 1.
>
> ---
>
> # 3 Preliminaries: Diffusion Feature
>
> Diffusion models consist of a neural network module and a diffusion scheduler.
> The network is an end-to-end network, which can be formally denoted as $\epsilon_\theta$, where $\theta$ is the parameters.
> The diffusion scheduler is the core of diffusion models.
> While previous generative models such as GAN expect to generate an image directly in one step, the diffusion scheduler turns the generation process into a timeline comprised of many timesteps.
> Generally, we use a **smaller / larger timestep** to indicate **less / more noises**, and $t=0$ / $t=T$ for clean images / total noises.
>
> Next, we will explain how convolutional features and attention features are extracted using a common pipeline for diffusion features, with the visual illustration in Figure 2.
> Given an input image $x\in\mathbb{R}^{3\times h \times w}$, where $h, w$ are height and width, the extraction process has the following steps:
> (i) A pre-trained VAE encodes the input image into the latent space, inducing $x_0\in\mathbb{R}^{4\times h' \times w'}$, as a common practice presented in \cite{rombach2022high, podell2023sdxl}.
> (ii) $x_0$ acts as the input of the forward diffusion process, *i.e.*, timestep $t=0$.
> As suggested in \cite{baranchuk2021label}, it is beneficial to extract diffusion features at non-zero timestep. Following this practice, we set $t=50$ and get $x_t$.
> (iii) $x_t$, along with the timestep $t$, a textual prompt $c$, is sent into the pre-trained diffusion UNet $\epsilon_\theta$, *i.e.*, $\epsilon_\theta(x_t,t,c)$.
> The generation techniques selected by our method are also applied here to modify $\theta, c$.
> (iv) Convolutional and attention features are gathered during the computation of the backbone.
> For convolutional features, we gather the output activations of each resolution in the upsampling stage of the UNet.
> For attention features, we obtain the mean value of the similarity maps between query and key in all cross-attention layers.

---

> ### Author Response · Authors · 2024-08-13
>
> (Part 2 of 4)
> ## 4.2 Cause of Content Shift
>
> After the existence of content shift is confirmed in diffusion features, we next aim to find its cause.
> Unlike more conventional feature extractors such as ResNet \cite{he2016deep}, the inputs to diffusion models are not the original image (Figure 4(a)), but its noisy version (Figure 4(b)), as enforced by the diffusion process \cite{ho2020denoising}.
> The early layers of diffusion models even further add more noise to the inner representations (Figure 4(c)).
> However, the diffusion UNet gains the ability from vast pre-training to reconstruct clean inner representations from noisy inputs (Figure 4(d)), roughly at the middle of the UNet structure.
> Additionally, the shortcut structures in UNet also help the reconstruction by passing some high-frequency details.
> Afterward, the diffusion UNet will further predict noises based on the reconstructed representations (Figure (e)).
>
> Notably, the diffusion features we are using are in fact the reconstructed representations, which answers why clean diffusion features can be obtained even though the inputs are noisy.
> Despite the reconstruction ability, however, many high-frequency details are potentially blurred out by input noises, and thus their reconstruction is mostly based on "imagination".
> This leads to possible drift from the original image during reconstruction \cite{Daras23consistent}.
> Naturally, the content shift phenomenon in extracted diffusion features reflects the drift during reconstruction.
> **Consequently, content shift is an inherent characteristic of diffusion models and diffusion features, which suggests its broad existence across models and timesteps**.
>
> ---
>
> ## 5.2 Quantitative Evaluation
>
> We also propose a quantitative metric for the evaluation of generation techniques.
> We set feature extracted at $t=0$ as reference $FEAT_{ref} \in \mathbb{R}^{c\times h\times w}$ as it is less affected by noises. We use the Laplacian operator to evaluate the contour difference between feature $FEAT$ and the reference:
>
> $$
> diff=\sum_{i,j}^{h\times w}|\left\|Laplacian(FEAT_{ref},i,j)\right\|_2-\left\|Laplacian(FEAT,i,j)\right\|_2|\in(-\infty,1]
> $$
>
> We then set a feature with stronger content shift as an anchor and compare $diff_{anchor}$ with other features.
>
> $$
>     Score=\frac{(diff_{anchor}-diff)}{diff_{anchor}}
> $$
>
> $Score=1$ means an exact match, and a smaller value indicates more shift.
> In this way, we can measure the extent of content shift in extracted features using different generation techniques and thus evaluate the suppression effect of techniques.
> Noticeably, the evaluation of this quantitative metric can be well approximated by the previously proposed qualitative evaluation, so we recommend the qualitative evaluation if efficiency is desired.

---

### Official Review · Reviewer_ECpW · 2024-07-08

**Soundness:** 4
**Presentation:** 3
**Contribution:** 3
**Rating:** 7
**Confidence:** 5

**Summary:**

This paper explores how diffusion models, typically used for generative tasks, can also serve discriminative purposes by using inner activations as features. The authors point out that these features often suffer from a phenomenon named content shift, i.e., the features are semantically different from the input image. This content shift negatively impacts the performance of diffusion features. The authors propose to use off-the-shelf generation techniques to suppress content shift. To this end, they introduce a practical guideline called GATE. Their approach, which is validated through extensive experiments, significantly improves performance on various tasks and datasets.

**Strengths:**

1) As far as I know, this paper is the first to reveal and systematically analyze the phenomenon of content shift in diffusion features, highlighting its universal and harmful effects on discriminative tasks.
2) The authors propose a practical guideline (GATE) that utilizes off-the-shelf generation techniques to suppress content shift. This method leverages existing tools and advancements in the field, providing a convenient approach to enhancing diffusion feature quality.
3) The effectiveness of the proposed approach is demonstrated through extensive experiments on various tasks and datasets, showing significant improvements over state-of-the-art methods.
4) The proposed solution is not limited to a specific task or model but is shown to be a generic booster for diffusion features. This broad applicability enhances the potential impact and utility of the approach in different scenarios and for various applications.

**Weaknesses:**

1) The authors propose the GATE guideline for evaluating generation techniques. Then, the actual implementation includes three successful techniques and a failed one. However, as pointed out by the authors, there are abundant techniques for generation. Hence, it is recommended to provide more failure cases to validate the negative impact of the content shift.
2) Some details can be further clarified. To be specific, which results are implemented by the authors or borrowed from previous works could be more explicitly stated. Besides, some results in Table 1 and Table 2 are missing. The authors should make a further explanation.
3) Besides, some typos and minor errors found in the paper:
- Section 2, Line 66: "diffusion feature" should be "diffusion features".
- Section 4.2, Line 136: "consist the diffusion features we extract" should be "constitute the diffusion features we extract".
- In reference [1], “ECCV Workshopt” should be “ECCV Workshop”.

**Questions:**

Please see the weakness.

**Limitations:**

Yes, the authors discuss their limitations in Section 7.

---

> ### Author Rebuttal · Authors · 2024-08-06
>
> Thanks for your constructive comments, and we would like to make the following response.
>
> > **Q1:**
> Hence, it is recommended to provide more failure cases to validate the negative impact of the content shift.
>
> **A1:**
> We have included another failure case, IP-Adapter, in the rebuttal.
> For details, please refer to the **global response GQ2**.
>
> > **Q2:**
> To be specific, which results are implemented by the authors or borrowed from previous works could be more explicitly stated. Besides, some results in Table 1 and Table 2 are missing. The authors should make a further explanation.
>
> **A2:**
> Thanks for your patience and advice. We will refine these details further and make some explanation here.
> In Table 1, Baseline for semantic correspondence and GATE for both tasks are implemented by us, and all the other results are borrowed from previous studies [1-4, 12, 23, 26, 30, 36, 39, 53].
> In Table 2, MaskCLIP and ODISE results are borrowed from previous studies [8, 45]. As they do not provide results on CityScapes and it is hard to extend the codes on this dataset, we do not have CityScapes results for the two methods. Both our GATE and VPD are implemented by us. Specifically, the original results reported by VPD are based on full-scale fine-tuning of diffusion UNet, which is not fair as we do not train the diffusion model in our setting. Therefore, we re-evaluated VPD with the diffusion UNet frozen and reported our results.
>
> > **Q3:**
> Besides, some typos and minor errors found in the paper.
>
> **A3:**
> Thanks for your patience. We will pay attention to these typos and errors during refinement.

---

> > ### Comment · Reviewer_ECpW · 2024-08-13
> >
> > Thanks for your detailed responses and most of my concerns are addressed. I maintain my score.

---

### Official Review · Reviewer_cofK · 2024-07-13

**Soundness:** 3
**Presentation:** 4
**Contribution:** 3
**Rating:** 7
**Confidence:** 5

**Summary:**

The authors focus on the task of suppressing content shift in diffusion features using off-the-shelf generation techniques. The authors observe that diffusion features, which are extracted from pre-trained diffusion models, suffer from a phenomenon called content shift, where there are content differences between the input image and the features extracted. The strength of the proposed method lies in utilizing off-the-shelf generation techniques to suppress this content shift, leveraging their ability to control the recovering process in diffusion models. Experimental validation shows that the proposed GATE framework effectively enhances diffusion features across various tasks and datasets, outperforming state-of-the-art methods.

**Strengths:**

(1) The authors identify that even minor, visually imperceptible content shifts can negatively impact the discriminative performance of diffusion features, highlighting the necessity to suppress this phenomenon to enhance feature quality.
(2) The method leverages off-the-shelf generation techniques to suppress content shift, which maintains computational efficiency and achieves significant improvements.
(3) Extensive experiments confirm that the method outperforms state-of-the-art methods in multiple benchmarks, validating its potential for widespread application.
(4) The paper is well organized, making it easy to understand.

**Weaknesses:**

(1) Some details can be more clarified. For example, the authors propose to use  fine-grained prompts to surpass content shift. I wonder how to design these prompts. Does it require us to design prompts for each input?
(2) The paper focuses on empirical validation and lacks a thorough theoretical analysis of why and how the proposed techniques effectively suppress content shift. It is recommended to provide more insights or explanations.

**Questions:**

See the weakness.

**Limitations:**

Yes.

---

> ### Author Rebuttal · Authors · 2024-08-06
>
> Thanks for your constructive comments, and we would like to make the following response.
>
> > **Q1:**
> I wonder how to design these prompts. Does it require us to design prompts for each input?
>
> **A1:**
> We observe the common characteristics of the training images and design a general prompt for all images.
> In this way, we do not need to design prompts for each input nor see the images of the testing set.
> More advanced methods can also be utilized, such as a pre-trained image captioner. We do not use image captioners in the manuscript, but some additional results taking this approach are listed in the **global response GQ3**.
>
> > **Q2:**
> The paper focuses on empirical validation and lacks a thorough theoretical analysis of why and how the proposed techniques effectively suppress content shift. It is recommended to provide more insights or explanations.
>
> **A2:**
> Thanks for this constructive suggestion! We focus this paper on enhancing the actual performance and thus leave thorough theoretical analysis to future work. Nevertheless, we have made some preliminary discoveries.
> As shown in Fig.4, from pre-training, diffusion models obtain the ability to reconstruct clean inner latent representations from noisy inputs. However, noises may make some details blurry, thus making the reconstruction drift from the input image, which leads to content shift in diffusion features.
> The three selected techniques can suppress content shift due to two factors.
> On one hand, they introduce additional information into this reconstruction process.
> On the other hand, the techniques are designed to steer the reconstruction towards the introduced information.
> The two factors combined together can guide the reconstructed latent representation towards the input image, thus suppressing content shift.

---

### Official Review · Reviewer_z9si · 2024-07-13

**Soundness:** 3
**Presentation:** 2
**Contribution:** 3
**Rating:** 4
**Confidence:** 3

**Summary:**

This paper addresses the issue of content shift in diffusion features, which are inner activations extracted from pre-trained diffusion models used for discriminative tasks. The content shift refers to the phenomenon where there are content differences between the features and the input images. In the manuscript, the authors showed the cause of content shift as an inherent characteristic of diffusion models, related to information drift during the image recovery process from noisy inputs. The proposed method utilizes off-the-shelf generation techniques to suppress content shift and introduces GATE for evaluating and implementing the techniques. The proposed method is simple yet novel.

**Strengths:**

+ The paper identifies and systematically analyzes the content shift in diffusion features, which has not been studied before. it is a simple but novel method.
+ The related work and the explanation of the phenomenon are clear. They supported the explanation with visual examples and empirical results.
+ They compared the proposed method with different state-of-the-art methods on different benchmark datasets to show the effectiveness of the proposed method.

**Weaknesses:**

- The paper lacks clarity in explaining the exact process of how attention features are extracted from the diffusion model. Figures 1 and 2, which are supposed to illustrate this process, do not provide enough detail.
- The details of the overall process of the proposed method are missing.
- The effectiveness of the GATE guideline is demonstrated using a limited set of generation techniques. A broader range of techniques should be evaluated to strengthen the claim of its general applicability.
- Even though the authors provided comparison experiments in the manuscript and ablation study in the supplementary materials, the results do not clearly demonstrate the superiority of the proposed method. More detailed and extensive comparative analyses with additional evaluation metrics and qualitative evaluation are needed to show the advantages of the proposed method.

**Questions:**

1. Please provide a more detailed explanation and additional diagrams to clarify how attention features are extracted from the diffusion model in Figure 1 and 2.
2. Please provide details of the overall process of the proposed method.
3. How does the proposed method perform on tasks involving different types of images or datasets with varying levels of noise and complexity?

**Limitations:**

1. The paper should improve the clarity of the proposed method, especially in explaining the extraction of features from the diffusion model. Providing a more detailed and step-by-step explanation would help readers understand the method better.
2. The authors should consider demonstrating the experimental results in a better way with additional evaluation metrics and extensive qualitative evaluation.

---

> ### Author Rebuttal · Authors · 2024-08-06
>
> Thanks for your constructive comments, and we would like to make the following response.
>
> > **Q1:**
> Please provide a more detailed explanation and additional diagrams to clarify how attention features are extracted from the diffusion model in Fig.1 and Fig.2.
>
> **A1:**
> Modern diffusion models use cross-attention between the latent image and prompts to enable user interaction. We extract the similarity scores between query (latent image) and key (prompt) and average them over all layers as attention features.
> For more details, please refer to the **global response GQ1** and the attached PDF.
>
> > **Q2:**
> Please provide details of the overall process of the proposed method.
>
> **A2:**
> Perhaps due to the omitted details of **Extract Features with Techniques** in Fig.7, the overall pipeline is a bit confusing. In fact, this step follows the practice in the prior arts [2]. We first encode an input image with VAE, then add noises to it, and finally send the image into diffusion UNet to extract features.
>
> For more details, please refer to the **global response GQ1** and the attached PDF.
>
> > **Q3:**
> A broader range of techniques should be evaluated to strengthen the claim of its general applicability.
>
> **A3:**
> We have also evaluated two techniques, classifier-free guidance and IP-Adapter. As the results presented in Appendix and **global response GQ2** show, the two techniques fail to suppress content shift and do not achieve significant performance gain. These results again strengthen the general applicability of our GATE guideline.
>
> > **Q4:**
> More detailed and extensive comparative analyses with additional evaluation metrics and qualitative evaluation are needed to show the advantages of the proposed method.
>
> **A4:**
> According to your suggestion, we have provided more comparative analyses with additional metrics, including mIoU, aAcc, and mAcc. Please refer to the **global response GQ4** for more details.
>
> For more qualitative results, besides the visualization in Fig.10, we also include additional feature visualization in the PDF attached to the global response. From the visualization, we can observe that:
>
> i) The attention features become clearer and closer to the input image when more generation techniques are applied according to GATE, showing the suppression effect on content shift.
> ii) Notably, for the second image where a person is riding a horse, the baseline attention feature fails to follow the instruction, i.e., attending only to the horse and ignoring the person. In contrast, generation techniques applied according to GATE help attention features attend to the correct object.
> iii) From convolutional features, we can see the application of generation techniques brings stronger diversity.
>
> > **Q5:**
> How does the proposed method perform on tasks involving different types of images or datasets with varying levels of noise and complexity?
>
> **A5:**
> Among the three generation techniques, only fine-grained prompts can be sensitive to image types and noise levels. For simplicity, we tackle this challenge via general prompts, whilst a pre-trained image captioner is also an option.
>
> For more details, please refer to the **global response GQ3**.

---

> > ### Comment · Reviewer_z9si · 2024-08-13
> >
> > Thank you for your detailed rebuttal and the clarifications provided. The additional explanations and diagrams are clear and helpful to clarify several aspects of the proposed method. The rebuttal have effectively addressed the concerned I raised. However, I want to mention that the authors' responses include substantial clarifications and additional analyses that may border on revisions rather than the typical rebuttal. I am not sure that this level of change aligns with NeurIPS policies on rebuttals.

---

> > > ### Author Response · Authors · 2024-08-14
> > >
> > > Dear Reviewer z9si,
> > >
> > > We greatly appreciate your diligence, expertise, and efforts!
> > >
> > > We are delighted to hear from you that our detailed rebuttal and clarifications have effectively addressed your concerns. Through the rebuttal and discussion, we pursue better ways to clarify some important concepts of our method (such as the feature extraction pipeline and some experimental details), while the core idea and main contributions stay unchanged. Perhaps due to the way of our writing, these concepts are not fully clarified in the original version. Furthermore, it seems that the rebuttal policies allow such revision for clarifications and analyses, according to the additional page for the camera-ready version.
> > >
> > > For your convenience, we attach the related policies right here.
> > >
> > > > Author responses: Authors will have one week to view and respond to initial reviews. Author responses may not contain any identifying information that may violate the double-blind reviewing policy. Authors may not submit revisions of their paper or supplemental material, but may post their responses as a discussion in OpenReview. This is to reduce the burden on authors to have to revise their paper in a rush during the short rebuttal period.
> > >
> > > > After the initial response period, authors will be able to respond to any further reviewer/AC questions and comments by posting on the submission’s forum page. The program chairs reserve the right to solicit additional reviews after the initial author response period.  These reviews will become visible to the authors as they are added to OpenReview, and authors will have a chance to respond to them.
> > >
> > > > The main text of a submitted paper is limited to nine content pages, including all figures and tables. Additional pages containing references don’t count as content pages. If your submission is accepted, you will be allowed an additional content page for the camera-ready version.

---

> ### Author Response · Authors · 2024-08-13
>
> Dear Reviewer z9si,
>
> As the rebuttal deadline approaches, could you please let us know if our newly added response has addressed your concerns?
> Thank you once again for your time and valuable feedback!

---

### Author Rebuttal · Authors · 2024-08-06

Dear SAC, AC, and reviewers,

Thanks for your valuable feedback. Based on your comments, we first offer a global response to some common questions.

> **GQ1:**
More details about the overall pipeline and how attention features are extracted.
**(Q1, Q2 of z9si, Q1 of RvKH)**

**GA1:**
Thanks for this constructive suggestion! In the original manuscript, we provided an overview of the proposed GATE framework in Fig.7 but did not detail the process of extracting attention features. According to this advice, the PDF attached to the global rebuttal details the extraction process, i.e., the top-right module **Extract Features with Techniques** in Fig.7. In the revised manuscript, we will replace Fig.2 in Preliminary section with the new figure.

Specifically, given an input image $x\in\mathbb{R}^{3\times h \times w}$, where $h, w$ are height and width, the extraction process has the following steps:

i) A pre-trained VAE encodes the input image into the latent space, inducing $x_0\in\mathbb{R}^{4\times h' \times w'}$, as a common practice presented in [31, 32].

ii) $x_0$ acts as the input of the forward diffusion process, i.e., timestep $t=0$. As suggested in [2], it is beneficial to extract diffusion features at non-zero timestep. Following this practice, we set $t=50$ and get $x_t$.

iii) $x_t$, along with the timestep $t$, a textual prompt $c$, is sent into the pre-trained diffusion UNet $\epsilon_\theta$, where $\theta$ denotes the UNet backbone, i.e., $\epsilon_\theta(x_t,t,c)$. The generation techniques are used here to modify $\theta, c$.

iv) Finally, diffusion features can be obtained by:
 - **Convolutional Feature**: The pre-trained diffusion UNet is comprised of multiple resolutions, which can be divided into downsampling stage, middle stage, and upsampling stage. We extract the output activations of each resolution in the upsampling stage as convolutional features.
 - **Attention Feature**: In modern diffusion models, there are many cross-attention layers, which calculate the attention between the latent image and the textual prompt. These cross-attention layers enable users to control the generated image content using natural language prompts. We extract the similarity maps between query (image) and key (prompt). Afterward, we average the maps over all layers and obtain an attention map for each token in the prompt. All these maps consist attention features.

> **GQ2:**
Evaluate more generation techniques.
**(Q3 of z9si, Q1 of ECpW, Q2 of RvKH)**

**GA2:**
Besides fine-grained prompt, ControlNet, and LoRA, we have also evaluated classifier-free guidance [15] and IP-Adapter [48]. According to our qualitative evaluation guideline, these two techniques fail to suppress content shift effectively. Furthermore, our quantitive results also show that they do not bring significant benefits, which again validates the effectiveness of our qualitative guideline.

To be specific, classifier-free guidance aims to enhance generation quality. However, better image quality does not necessarily suppress content shift. We have included this technique in the Appendix as a failure case study.

IP-Adapter shares similar architecture with  ControlNet. However, its goal is image variation, instead of strict control. Hence, it is less effective on mitigating content shift than ControlNet. To support this claim, we include some ablation experimental data below, which is conducted on the Horse-21 dataset.

|Method|mIoU|
|:--|:-:|
|DDPM|65.0 ± 0.8|
|GATE (Only IP-Adapter)|65.8 ± 1.3|
|GATE (Only ControlNet)|66.2 ± 1.2|
|GATE (Full)|**67.2 ± 1.1**|

In conclusion, if a generation technique can make the generation result closer to the additional information introduced to the generation process, the technique tends to be helpful. This property can be efficiently evaluated using our GATE guideline for more techniques.

> **GQ3:**
How to apply the selected techniques to datasets with various types of images and noise levels?
**(Q5 of z9si, Q1 of cofK, Q7 of RvKH)**

**GA3:**
Among the three selected techniques, ControlNet and LoRA are essentially insensitive to image types, noise levels, and complexity. Hence, they can be applied to various types of images and datasets.

Fine-grained prompts can be sensitive to these factors if we use image-specific prompts. To tackle this challenge, we have two solutions: i) a relatively general prompt or ii) a pre-trained image captioner to generate specific prompts for each image. For simplicity, the manuscript chooses the first approach for the ADE20K dataset, which contains images of varying types. The corresponding performance gain shows that GATE with general prompts can apply to this scenario. Additionally, according to this suggestion, we provide experimental results of the second approach below, using captioner Kosmos-2 [29]. We can find that GATE (Captioner) achieves comparable performance to GATE (Standard). Hence, we choose general prompts in practice for simplicity and efficiency.

|Method|mIoU|
|:--|:-:|
|DDPM|65.0 ± 0.8|
|GATE (Captioner)|**67.3 ± 1.4**|
|GATE (Standard)|67.2 ± 1.1|

> **GQ4:**
More comparative experiments are needed. **(Q4 of z9si, Q3, Q8 of RvKH)**

**GA4:**
For additional evaluation metrics, we report the mIoU, aAcc and mAcc results on ADE20K and CityScapes.
- mIoU is as defined in the manuscript;
- aAcc is the classification accuracy of all pixels;
- mAcc averages the accuracy over categories.

We also report image classification results, where Baseline is diffusion feature without GATE.

These results again validate the effectiveness of GATE.

- ADE20K
|Method|mIoU|aAcc|mAcc|
|:-|:-:|:-:|:-:|
|VPD|37.63|79.16|50.08|
|GATE|40.51|79.68|54.90|

- CityScapes
|Method|mIoU|aAcc|mAcc|
|:-|:-:|:-:|:-:|
|VPD|55.06|90.14|68.96|
|GATE|64.20|92.83|76.98|

- CIFAR10
|Method|Acc(\%)|
|:-|:-:|
|ResNet-50 [a]|93.62|
|Baseline|94.55|
|GATE|95.21|

[a] https://github.com/kuangliu/pytorch-cifar (5.9k star)

---

### Author Response · Authors · 2024-08-07

We provide the full-scale experimental details here.

We extract features at $t=50$. When ControlNet is applied, except for ADE20K and CityScapes, we additionally start multi-step denoising from $t=60$.
- **Semantic Correspondence**
	- Convolutional features: two features per setting under different random seeds:
		- Two with fine-grained prompt "a photo of x, high quality, best quality, highly realistic, masterpiece, high resolution", where "x" is the concatenation of all class labels.
		- Two with LoRA.
		- Two with ControlNet.
	- Attention features are obtained using a prompt including all object categories, with ControlNet (denoising from $t=60$) and LoRA also applied.
- **Label-Scarce Segmentation: Horse-21**
	- Convolutional features extracted at $t=50$:
		- A feature using fine-grained prompt "a horse, high quality, best quality, highly realistic, masterpiece".
		- A feature using ControlNet, with a simple prompt "a horse".
		- Two features using different LoRA weights and a simple prompt. The first LoRA weight is trained to generate high-quality images, while the second LoRA weight is trained until it slightly overfits.
	- Attention features are obtained using a prompt "a photo of a single horse running in a sports field, with a well-equipped rider on its back, seems they are in a competition, high quality, best quality, highly realistic, masterpiece", with ControlNet and the first LoRA weight applied.
	- Additionally, we concatenate the output feature of amalgamation with a feature extracted using DDPM [2].
- **Label-Scarce Segmentation: Bedroom-28**
	- Convolutional features extracted at $t=50$:
		- A feature using fine-grained prompt "a photo of a tidy and well-designed bedroom". It is found that quality prompts such as "high quality" will be interpreted as the quality of the room instead of the image, so such prompts are not utilized.
		- A feature using ControlNet, with a simple prompt "a bedroom".
		- One feature using LoRA weight, which is moderately trained to generate high-quality images.
	- Attention features are obtained using a prompt "bedroom, a bed, some bedroom furniture, lights, a door, ceiling, floor, walls, pillow, quilt, chair and table, window, in good quality". ControlNet and LoRA are also applied.
	- Additionally, we concatenate the output feature of amalgamation with a feature extracted using DDPM [2].
- **Semantic Segmentation: ADE20K**
	- Convolutional features extracted at $t=50$:
		- One feature using only fine-grained prompt "a highly realistic photo of the real world. It can be an indoor scene, or an outdoor scene, or a photo of nature. high quality" plus all category labels.
		- Two features using the same prompt as above and additionally ControlNet and LoRA. The two features use two different LoRA weights, where one is moderately trained and the other slightly overfits.
	- No attention features are used. We find that in very complex scenes attention features can be of low quality and bring slight degradation instead of enhancement.
- **Semantic Segmentation: CityScapes**
	- Convolutional features extracted at $t=50$:
		- One feature using only fine-grained prompt "An urban street scene with multiple lanes, various buildings, traffic lights, cars in the lanes, and pedestrians, highly realistic".
		- Two features using the same prompt as above and additionally ControlNet and LoRA. The two features use two different LoRA weights, where one is moderately trained and the other slightly overfits.
	- No attention features are used, based on the same observation as for ADE20K.

---

### Comment · Area_Chair_8K43 · 2024-08-10
**Please respond to the authors' rebuttal.**

Dear Reviewers,

The authors have posted their rebuttals to the reviews. Could you please respond to the rebuttals?
Please engage in the discussion with the authors. Your help is much appreciated.

Thanks,

AC

---

### Decision · Program_Chairs · 2024-09-25

**Decision:**

Accept (poster)

**Comment:**

This paper is about suppressing the content shift in recently popular "diffusion features" (using pre-trained diffusion models to retrieve feature representations). The phenomenon of content shift is newly discovered by the authors, i.e., the content of the feature representation shifts from the original input due to the "noisy" nature of diffusion models. To resolve this problem, the authors propose to utilize popular conditioning techniques (e.g., prompt-based, ControlNet, and LoRA) in a very unusual but clever way: conditioning the features based on the very original image. This is very simple but quite effective, achieving state-of-the-art performance with substantial improvements. The proposed method is also quite efficient in that off-the-shelf techniques can be directly used, which also philosophically conforms with the recent trends of AI. All reviewers have agreed on the potential impact of the topic itself and the proposed method.

However, there was a not-so-small issue during the author-reviewer discussion: The original submission lacked details and was written at a higher level, so a large number of additional details were submitted as a rebuttal. This amount concerned some of the reviewers, and in response, the authors have posted a partial revision of the paper as another rebuttal. This is not allowed in the NeurIPS policy, and there was a heated discussion between the authors and the reviewers regarding its appropriacy. After a discussion between AC and SAC, it was decided that the reviewers should ignore the revision and continue the discussion with AC. The issue still resided in that the amount of revision needed is quite substantial.

After the discussion, the AC personally reviewed the paper in detail to yield the following conclusion: The required revision is mostly about details and not the core contribution, as the authors argued. The method is largely based on very well-known off-the-shelf techniques (diffusion features, conditioning techniques like prompt-based, ControlNet, and LoRA), and the only novel twist introduced in the paper is simply changing the conditioning input to an unusual one (the original input itself), which is in fact not quite different from the usual ones (some other input to steer the generated result) in characteristics. The details needed here are more about tuning issues rather than deal-breaking design issues. Accordingly, the required revision is not too substantial (in a qualitative sense), considering the merit of the topic/message and the effectiveness/efficiency of the proposed method.

That being said, the additional details discussed in the rebuttal are indeed required and must definitely be added to the final camera-ready version. Under this strict condition, an accept decision is recommended.